# DYNAMICCITY: LARGE-SCALE 4D OCCUPANCY GENERATION FROM DYNAMIC SCENES

**Hengwei Bian**[1,2,∗] **Lingdong Kong**[1,3] **Haozhe Xie**[4] **Liang Pan**[1,†,‡] **Yu Qiao**[1] **Ziwei Liu**[4]

[1]Shanghai AI Laboratory  [2]Carnegie Mellon University  [3]National University of Singapore
[4]S-Lab, Nanyang Technological University

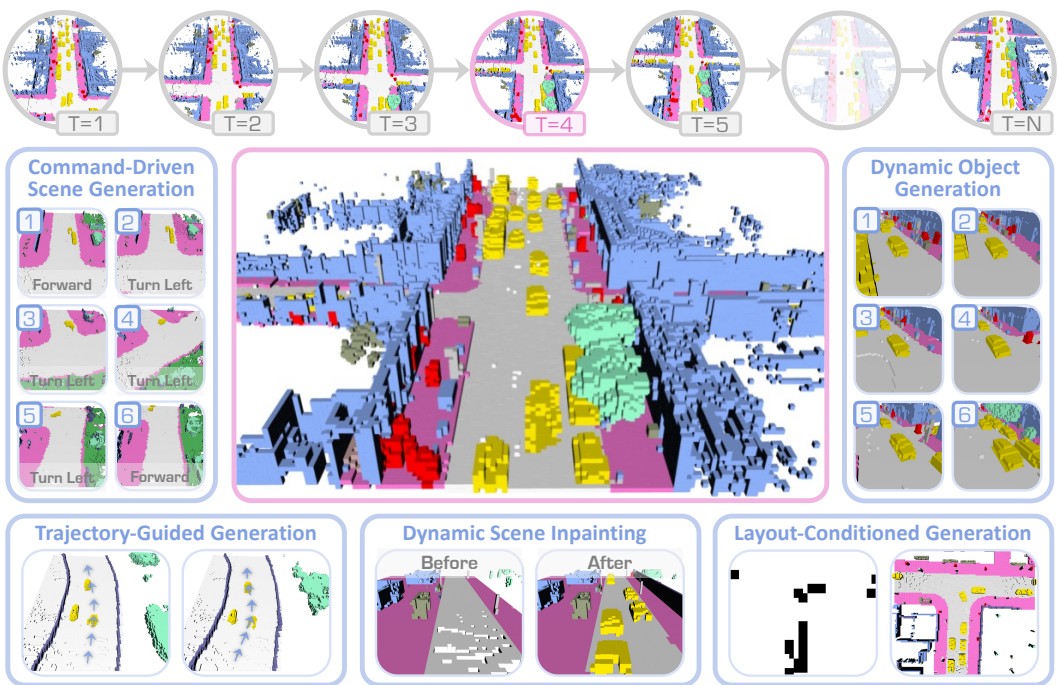

Figure 1: **Dynamic occupancy generation from** DynamicCity**.** We introduce a new generation model that generates diverse 4D scenes of large spatial scales ($80 \times 80 \times 6.4$ meter$^3$) and long sequential modeling (up to $128$ frames), enabling a diverse set of downstream applications. For more detailed examples, kindly refer to our **Project Page:** **https://dynamic-city.github.io**.

## ABSTRACT

Urban scene generation has been developing rapidly recently. However, existing methods primarily focus on generating static and single-frame scenes, overlooking the inherently dynamic nature of real-world driving environments. In this work, we introduce DynamicCity, a novel 4D occupancy generation framework capable of generating large-scale, high-quality dynamic 4D scenes with semantics. DynamicCity mainly consists of two key models. **1)** A VAE model for learning HexPlane as the compact 4D representation. Instead of using naive averaging operations, DynamicCity employs a novel **Projection Module** to effectively compress 4D features into six 2D feature maps for HexPlane construction, which significantly enhances HexPlane fitting quality (up to **12.56** mIoU gain). Furthermore, we utilize an **Expansion & Squeeze Strategy** to reconstruct 3D feature volumes in parallel, which improves both network training efficiency and reconstruction accuracy than naively querying each 3D point (up to **7.05** mIoU gain, **2.06x** training speedup, and **70.84**% memory reduction). **2)** A DiT-based diffusion model for HexPlane generation. To make HexPlane feasible for DiT generation, a **Padded Rollout Operation** is proposed to reorganize all six feature planes of the Hex-Plane as a squared 2D feature map. In particular, various conditions could be introduced in the diffusion or sampling process, supporting **versatile 4D genera-**

---

∗Work done during an internship at Shanghai AI Laboratory.  † Corresponding author.  ‡ Project lead.

**tion applications**, such as trajectory- and command-driven generation, inpainting, and layout-conditioned generation. Extensive experiments on the CarlaSC and Waymo datasets demonstrate that DynamicCity significantly outperforms existing state-of-the-art 4D occupancy generation methods across multiple metrics. The code and models have been released to facilitate future research.

# 1 INTRODUCTION

Urban scene generation has garnered growing attention recently, which could benefit various related applications, such as robotics and autonomous driving. Compared to its 3D object generation counterpart, generating urban scenes remains an under-explored field, with many new research challenges such as the presence of numerous moving objects, large-scale scenes, and long temporal sequences (Huang et al., 2021). For example, in autonomous driving scenarios, a scene typically comprises multiple objects from various categories, such as vehicles, pedestrians, and vegetation, captured over a long sequence (*e.g.*, 200 frames) spanning a large area (*e.g.*, $80 \times 80 \times 6.4$ meters$^3$). Although in its early stage, 4D occupancy generation holds great potential to enhance the understanding of the 3D world, with wide-reaching and profound implications.

Due to the complexity of occupancy data, many efficient learning frameworks have been introduced for large-scale 3D scene generation. $\mathcal{X}^3$ (Ren et al., 2024b) utilizes a hierarchical voxel diffusion model to generate outdoor 3D scenes based on VDB data structure. PDD (Liu et al., 2023a) introduces a pyramid discrete diffusion model to progressively generate high-quality 3D scenes. SemCity (Lee et al., 2024) resolves outdoor scene generation by leveraging a triplane diffusion model. Despite achieving impressive occupancy generation, they primarily focus on generating static and single-frame 3D occupancy, and hence fail to effectively capture the dynamic nature of outdoor environments. Recently, a few works (Zheng et al., 2024b; Wang et al., 2024) have explored 4D scene generation. However, generating high-quality long-sequence 4D scenes is still a challenging and open problem (Nakashima & Kurazume, 2021; Nakashima et al., 2023).

In this work, we propose a novel 4D occupancy generation framework, DynamicCity, enabling generating large-scale, high-quality dynamic occupancy scenes, which mainly consists of two stages: **1)** a VAE network for learning compact 4D representations, *i.e.*, HexPlanes (Cao & Johnson, 2023; Fridovich-Keil et al., 2023); **2)** a HexPlane Generation model based on DiT (Peebles & Xie, 2023).

**VAE for 4D Occupancy.** Given a set of 4D occupancy scenes, DynamicCity first encodes the scene as a 3D feature volume sequence with a 3D backbone. Afterward, we propose a novel **Projection Module** based on transformer operations to compress the feature volume sequence into six 2D feature maps. In particular, the proposed projection module significantly enhances HexPlane fitting performance, offering an improvement of up to **12.56% mIoU** compared to conventional averaging operations. After constructing the HexPlane based on the projected six feature planes, we employ an **Expansion & Squeeze Strategy** (**ESS**) to decode the HexPlane into multiple 3D feature volumes in parallel. Compared to individually querying each point, ESS further improves HexPlane fitting quality (with up to **7.05%** mIoU gain), significantly accelerates training speed (by up to **2.06x**), and substantially reduces memory usage (by up to a relative **70.84%** memory reduction).

**DiT for HexPlane.** Using the encoded HexPlane, we use a DiT-based framework for generating HexPlane, enabling 4D occupancy generation. Training a DiT with token sequences naively generated from HexPlane may not achieve optimal quality, as it could overlook spatial and temporal relationships among tokens. Therefore, we introduce the **Padded Rollout Operation** (**PRO**), which reorganizes the six feature planes into a square feature map, providing an efficient way to model both spatial and temporal relationships within the token sequence. Leveraging the DiT framework, DynamicCity seamlessly incorporates various conditions to guide the 4D generation process, enabling **a wide range of applications** including hexplane-conditional generation, trajectory-guided generation, command-driven scene generation, layout-conditioned generation, and dynamic scene inpainting.

Our contributions can be summarized as follows:

- We propose DynamicCity, a high-quality, large-scale 4D occupancy generation framework consisting of a tailored VAE for HexPlane fitting and a DiT-based network for HexPlane generation, which supports various downstream applications.

- In the VAE architecture, DynamicCity employs a novel Projection Module to benefit in encoding 4D scenes into compact HexPlanes, significantly improving HexPlane fitting quality. Following, an Expansion & Squeeze Strategy is introduced to decode the HexPlanes for reconstruction, which improves both fitting efficiency and accuracy.
- Building on fitted HexPlanes, we design a Padded Rollout Operation to reorganize HexPlane features into a masked 2D square feature map, enabling compatibility with DiT training.
- Extensive experimental results demonstrate that DynamicCity achieves significantly better 4D reconstruction and generation performance than previous SoTA methods across **all** evaluation metrics, including generation quality, training speed, and memory usage.

## 2 RELATED WORK

**3D Object Generation** has been a central focus in machine learning, with diffusion models playing a significant role in generating realistic 3D structures. Many techniques utilize 2D diffusion mechanisms to synthesize 3D outputs, covering tasks like text-to-3D object generation (Ma et al., 2024), image-to-3D transformations (Wu et al., 2024a), and 3D editing (Rojas et al., 2024). Meanwhile, recent methods bypass the reliance on 2D intermediaries by generating 3D outputs directly in three-dimensional space, utilizing explicit (Alliegro et al., 2023), implicit (Liu et al., 2023b), triplane (Wu et al., 2024b), and latent representations (Ren et al., 2024b). Although these methods demonstrate impressive 3D object generation, they primarily focus on small-scale, isolated objects rather than large-scale, scene-level generation (Hong et al., 2024; Xie et al., 2025b). This limitation underscores the need for methods capable of generating complete 3D scenes with complex spatial relationships.

**Urban Scene Generation** extends the scope to larger, more complex environments. Earlier works used VQ-VAE (Zyrianov et al., 2022) and GAN-based models (Caccia et al., 2019; Nakashima et al., 2023) to generate LiDAR scenes. However, recent advancements have shifted towards diffusion models (Xiong et al., 2023; Ran et al., 2024; Nakashima & Kurazume, 2024; Zyrianov et al., 2022; Hu et al., 2024; Nunes et al., 2024), which better handle the complexities of expansive outdoor scenes. For example, (Lee et al., 2024) utilize voxel grids to represent large-scale scenes but often face challenges with empty spaces like skies and fields. While some recent works incorporate temporal dynamics to extend single-frame generation to sequences (Zheng et al., 2024b; Wang et al., 2024), they often lack the ability to fully capture the dynamic nature of 4D environments. Thus, these methods typically remain limited to short temporal horizons or struggle with realistic dynamic object modeling, highlighting the gap in generating high-fidelity 4D scenes.

**4D Generation** represents a leap forward, aiming to capture the temporal evolution of scenes. Prior works often leverage video diffusion models (Singer et al., 2022; Blattmann et al., 2023) to generate dynamic sequences (Singer et al., 2023), with some extending to multi-view (Shi et al., 2023; Xie et al., 2025a) and single-image settings (Rombach et al., 2022) to enhance 3D consistency. In the context of video-conditional generation, approaches such as (Jiang et al., 2023; Ren et al., 2023; 2024a) incorporate image priors for guiding generation processes. While these methods capture certain dynamic aspects, they lack the ability to generate long-term, high-resolution 4D scenes with versatile applications. Our method, DynamicCity, fills this gap by introducing a novel 4D generation framework that efficiently captures large-scale dynamic environments, supports diverse generation tasks (*e.g.*, trajectory-guided (Bahmani et al., 2024), command-driven generation), and offers substantial improvements in scene fidelity and temporal modeling.

## 3 PRELIMINARIES

**HexPlane** (Cao & Johnson, 2023; Fridovich-Keil et al., 2023) is an explicit and structured representation designed for efficient modeling of dynamic 3D scenes, leveraging feature planes to encode spacetime data. A dynamic 3D scene is represented as six 2D feature planes, each aligned with one of the major planes in the 4D spacetime grid. These planes are represented as $\mathcal{H} = [\mathcal{P}_{xy}, \mathcal{P}_{xz}, \mathcal{P}_{yz}, \mathcal{P}_{tx}, \mathcal{P}_{ty}, \mathcal{P}_{tz}]$, comprising a Spatial TriPlane (Chan et al., 2022) with $\mathcal{P}_{xy}$, $\mathcal{P}_{xz}$, and $\mathcal{P}_{yz}$, and a Spatial-Time TriPlane with $\mathcal{P}_{tx}$, $\mathcal{P}_{ty}$, and $\mathcal{P}_{tz}$. To query the HexPlane at a point $\mathbf{p} = (t, x, y, z)$, features are extracted from the corresponding coordinates on each of the six planes and fused into a comprehensive representation. This fused feature vector is then passed through a lightweight network to predict scene attributes for $\mathbf{p}$.

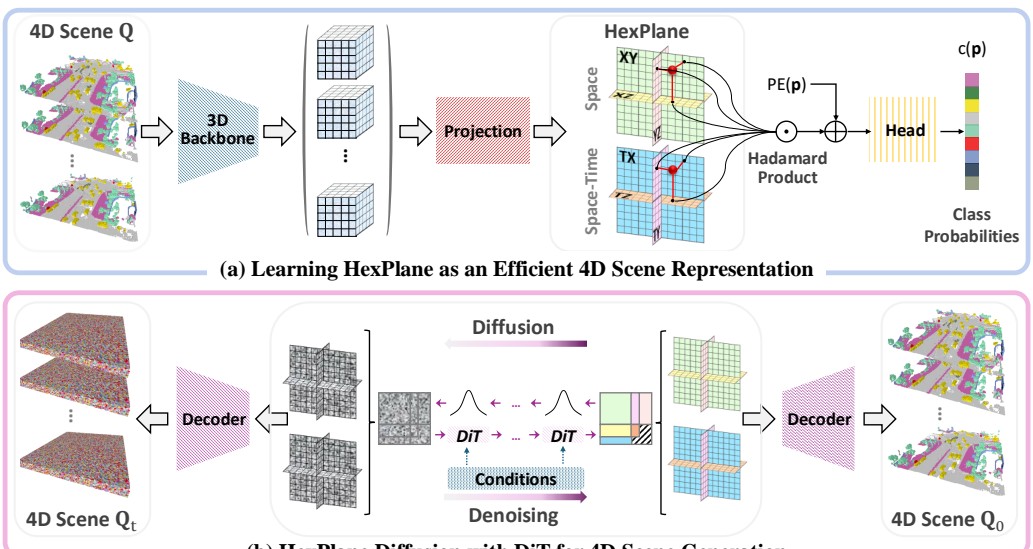

Figure 2: **Pipeline of dynamic scene generation.** Our DynamicCity framework consists of two key procedures: **(a)** Encoding HexPlane with an VAE architecture (*cf.* Sec. 4.1), and **(b)** 4D Scene Generation with HexPlane DiT (*cf.* Sec. 4.2).

**Diffusion Transformers** (DiT) (Peebles & Xie, 2023) are diffusion-based generative models using transformers to gradually convert Gaussian noise into data samples through denoising steps. The forward diffusion adds Gaussian noise over time, with a noised sample at step $t$ given by $\mathbf{x}_t = \sqrt{\alpha_t}\mathbf{x}_0 + \sqrt{1 - \alpha_t}\epsilon$, $\epsilon \sim \mathcal{N}(\mathbf{0}, \mathbf{I})$, where $\alpha_t$ controls the noise schedule. The reverse diffusion, using a neural network $\epsilon_\theta$, aims to denoise $\mathbf{x}_t$ to recover $\mathbf{x}_0$, expressed as: $\mathbf{x}_{t-1} = \frac{1}{\sqrt{\alpha_t}}(\mathbf{x}_t - \sqrt{1 - \alpha_t}\epsilon_\theta(\mathbf{x}_t, t))$. New samples are generated by repeating this reverse process.

## 4 OUR APPROACH

DynamicCity strives to generate dynamic 3D occupancy with semantic information, which mainly consists of a VAE for 4D occupancy encoding using HexPlane (Cao & Johnson, 2023; Fridovich-Keil et al., 2023) (Sec. 4.1), and a DiT for HexPlane generation (Sec. 4.2). Given a 4D scene, *i.e.*, a dynamic 3D occupancy sequence $\mathbf{Q} \in \mathbb{R}^{T \times X \times Y \times Z \times C}$, where $T$, $X$, $Y$, $Z$, and $C$ denote the sequence length, height, width, depth, and channel size, respectively, the VAE first aims to encode an efficient 4D representation, HexPlane $\mathcal{H} = [\mathcal{P}_{xy}, \mathcal{P}_{xz}, \mathcal{P}_{yz}, \mathcal{P}_{tx}, \mathcal{P}_{ty}, \mathcal{P}_{tz}]$, which is then decoded for reconstructing 4D scenes with semantics. After obtaining HexPlane embeddings, DynamicCity leverages a DiT-based framework for 4D occupancy generation. Diverse conditions could be introduced into the generation process, facilitating a range of downstream applications (Sec. 4.3). The overview of the proposed DynamicCity pipeline is illustrated in Fig. 2.

### 4.1 VAE FOR 4D OCCUPANCY

**Encoding HexPlane.** As shown in Fig. 3, the VAE could encode 4D occupancy $\mathbf{Q}$ as a HexPlane $\mathcal{H}$. It first utilizes a shared 3D convolutional feature extractor $\boldsymbol{f}_\theta(\cdot)$ to extract and downsample features from each occupancy frame, resulting in a feature volume sequence $\mathcal{X}_{txyz} \in \mathbb{R}^{T \times X \times Y \times Z \times C}$.

To encode and compress $\mathcal{X}_{txyz}$ into compact 2D feature maps of $\mathcal{H}$, we propose a novel Projection Module with multiple projection networks $\boldsymbol{h}(\cdot)$. To project a high-dimensional feature input $\mathcal{X}_{\text{in}} \in \mathbb{R}^{D_k^1 \times D_k^2 \times \cdots \times D_k^n \times D_r^1 \times D_r^2 \times \cdots \times D_r^m \times C}$ as a lower-dimensional feature output $\mathcal{X}_{\text{out}} \in \mathbb{R}^{D_k^1 \times D_k^2 \times \cdots \times D_k^n \times C}$, the projection network $\boldsymbol{h}_{S_r}(\cdot)$ first reshapes $\mathcal{X}_{in}$ into a 3-dimensional feature $\mathcal{X}'_{S_k S_r} \in \mathbb{R}^{S_k \times S_r \times C}$ by grouping the dimensions into the two new dimensions, *i.e.*, $S_k$ the dimension that will be kept, and $S_r$ the dimension that will be reduced, where $S_k = D_k^1 \times D_k^2 \times \cdots \times D_k^n$, and $S_r = D_r^1 \times D_r^2 \times \cdots \times D_r^m$. Afterward, $\boldsymbol{h}_{S_r}(\cdot)$ utilizes a transformer-based operation to project the

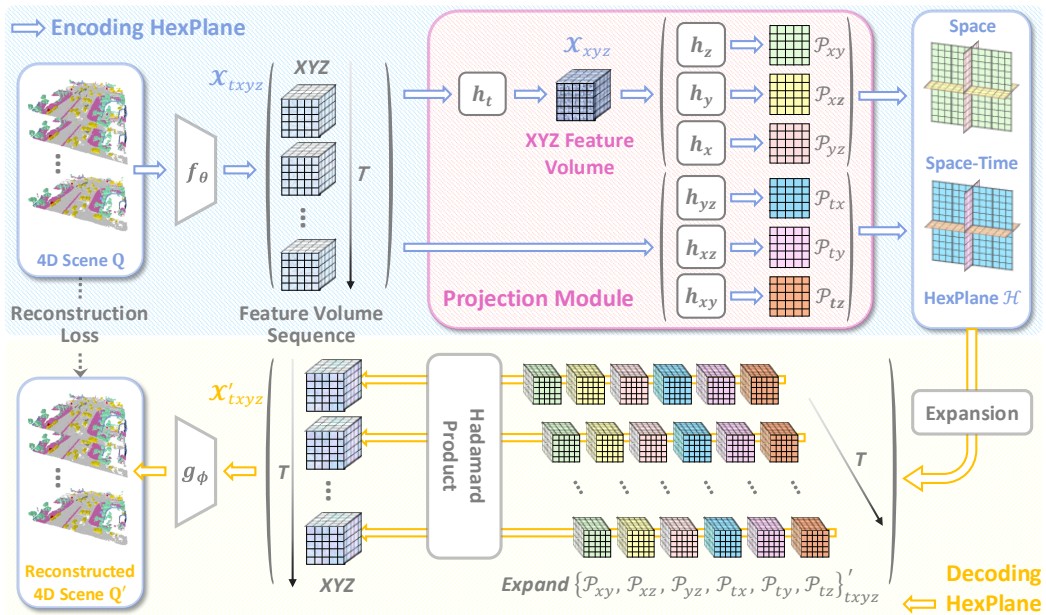

Figure 3: **VAE for encoding dynamic scenes.** We use HexPlane $\mathcal{H}$ as the 4D representation. $f_\theta$ and $g_\phi$ are convolution-based networks with downsampling and upsampling operations, respectively. $h(\cdot)$ denotes the projection network based on transformer modules.

reshaped feature $\mathcal{X}'_{S_k S_r}$ to $\mathcal{X}''_{S_k} \in \mathbb{R}^{S_k \times C}$, which is then reshaped to the expected lower-dimensional feature output $\mathcal{X}_{\text{out}}$. Formally, the projection network is formulated as:

$$\mathcal{X}_{\text{out}}^{\{D_k^1 \times D_k^2 \times \cdots \times D_k^n\} \times C} = \boldsymbol{h}_{S_r}(\mathcal{X}_{\text{in}}^{\{D_k^1 \times D_k^2 \times \cdots \times D_k^n\} \times \{D_r^1 \times D_r^2 \times \cdots \times D_r^m\} \times C}), \tag{1}$$

where their feature dimensions are added as the upscript for $\mathcal{X}^{\text{in}}$ and $\mathcal{X}^{\text{out}}$, respectively.

To construct the spatial feature planes $\mathcal{P}_{xy}$, $\mathcal{P}_{xz}$, and $\mathcal{P}_{yz}$, the Projection Module first generates the XYZ Feature Volume $\mathcal{X}_{xyz} = \boldsymbol{h}_t(\mathcal{X}_{txyz})$. Rather than directly access the heavy feature volume sequence $\mathcal{X}_{txyz}$, $\boldsymbol{h}_z(\cdot)$, $\boldsymbol{h}_y(\cdot)$, and $\boldsymbol{h}_x(\cdot)$ are applied to $\mathcal{X}_{xyz}$ for reducing the spatial dimensions of $\mathcal{X}_{xyz}$ along the z-axis, y-axis, and x-axis, respectively. The temporal feature planes $\mathcal{P}_{tx}, \mathcal{P}_{ty}$, and $\mathcal{P}_{tz}$ are directly obtained from $\mathcal{X}_{txyz}$ by simultaneously removing two spatial dimensions with $\boldsymbol{h}_{zy}(\cdot)$, $\boldsymbol{h}_{xz}(\cdot)$, and $\boldsymbol{h}_{xy}(\cdot)$, respectively. Consequently, we could construct the HexPlane $\mathcal{H}$ based on the encoded six feature planes, including $\mathcal{P}_{xy}, \mathcal{P}_{xz}, \mathcal{P}_{yz}, \mathcal{P}_{tx}, \mathcal{P}_{ty}$, and $\mathcal{P}_{tz}$.

**Decoding HexPlane.** Based on the HexPlane $\mathcal{H} = [\mathcal{P}_{xy}, \mathcal{P}_{xz}, \mathcal{P}_{yz}, \mathcal{P}_{tx}, \mathcal{P}_{ty}, \mathcal{P}_{tz}]$, we employ an **E**xpansion & **S**queeze **S**trategy (ESS), which could efficiently recover the feature volume sequence by decoding the feature planes in parallel for 4D occupancy reconstruction. ESS first duplicates and expands each feature plane $\mathcal{P}$ to match the shape of $\mathcal{X}_{txyz}$, resulting in the list of six feature volume sequences: $\{\mathcal{X}_{txyz}^{P_{xy}}, \mathcal{X}_{txyz}^{P_{xz}}, \mathcal{X}_{txyz}^{P_{yz}}, \mathcal{X}_{txyz}^{P_{tx}}, \mathcal{X}_{txyz}^{P_{ty}}, \mathcal{X}_{txyz}^{P_{tz}}\}$. Afterward, ESS squeezes the corresponding six expanded feature volumes with Hadamard Product:

$$\mathcal{X}'_{txyz} = \prod_{\text{Hadamard}} \{\mathcal{X}_{txyz}^{P_{xy}}, \mathcal{X}_{txyz}^{P_{xz}}, \mathcal{X}_{txyz}^{P_{yz}}, \mathcal{X}_{txyz}^{P_{tx}}, \mathcal{X}_{txyz}^{P_{ty}}, \mathcal{X}_{txyz}^{P_{tz}}\}. \tag{2}$$

Subsequently, the convolutional network $g_\phi(\cdot)$ is employed to upsample the volumes for generating dense semantic predictions $\mathbf{Q}'$:

$$\mathbf{Q}' = g_\phi(\texttt{Concat}(\mathcal{X}'_{txyz}, \texttt{PE}(\texttt{Pos}(\mathcal{X}'_{txyz})))), \tag{3}$$

where $\texttt{Concat}(\cdot)$ and $\texttt{PE}(\cdot)$ denote the concatenation and sinusoidal positional encoding, respectively. $\texttt{Pos}(\cdot)$ returns the 4D position $\mathbf{p}$ of each voxel within the 4D feature volume $\mathcal{X}'_{txyz}$.

**Optimization.** The VAE is trained with a combined loss $\mathcal{L}_{\text{VAE}}$, including a cross-entropy loss, a Lovász-softmax loss (Berman et al., 2018), and a Kullback-Leibler (KL) divergence loss:

$$\mathcal{L}_{\text{VAE}} = \mathcal{L}_{\text{CE}}(\mathbf{Q}, \mathbf{Q}') + \alpha \mathcal{L}_{\text{Lov}}(\mathbf{Q}, \mathbf{Q}') + \beta \mathcal{L}_{\text{KL}}(\mathcal{H}, \mathcal{N}(\mathbf{0}, \mathbf{I})), \tag{4}$$

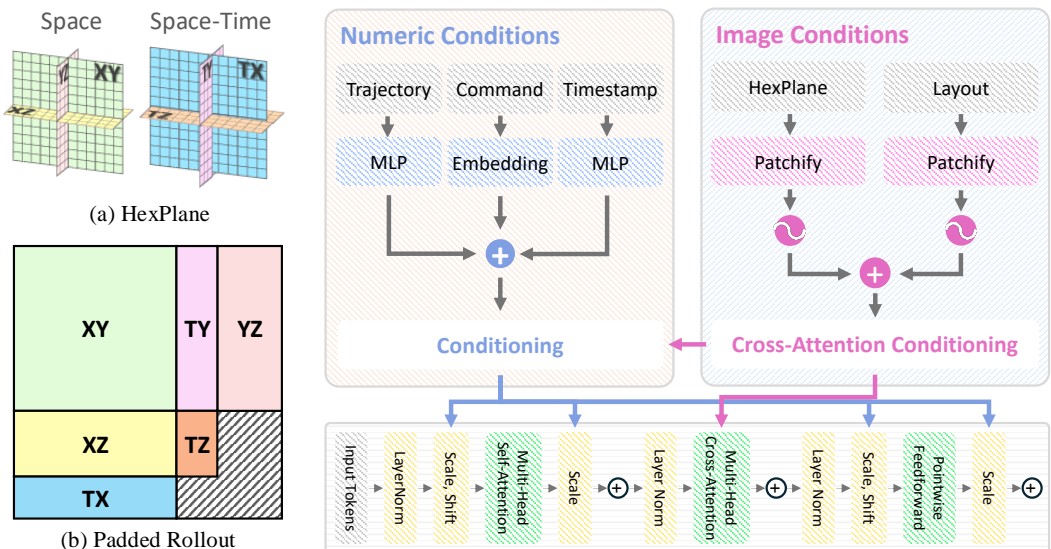

(a) HexPlane

(b) Padded Rollout

Figure 4: Padded Rollout

Figure 5: Condition Injection for DiT

where $\mathcal{L}_{\mathrm{CE}}$ is the cross-entropy loss between the input $\mathbf{Q}$ and prediction $\mathbf{Q}'$, $\mathcal{L}_{\mathrm{Lov}}$ is the Lovász-softmax loss, and $\mathcal{L}_{\mathrm{KL}}$ represents the KL divergence between the latent representation $\mathcal{H}$ and the prior Gaussian distribution $\mathcal{N}(\mathbf{0}, \mathbf{I})$. Note that the KL divergence is computed for each feature plane of $\mathcal{H}$ individually, and the term $\mathcal{L}_{\mathrm{KL}}$ refers to the combined divergence over all six planes.

## 4.2 DIFFUSION TRANSFORMER FOR HEXPLANE

After training the VAE, 4D semantic scenes can be embedded as HexPlane $\mathcal{H} = [\mathcal{P}_{xy}, \mathcal{P}_{xz}, \mathcal{P}_{yz}, \mathcal{P}_{tx}, \mathcal{P}_{ty}, \mathcal{P}_{tz}]$. Building upon $\mathcal{H}$, we aim to leverage a DiT (Peebles & Xie, 2023) model $D_\tau$ to generate novel HexPlane, which could be further decoded as novel 4D scenes (see Fig. 2(b)). However, training a DiT using token sequences naively generated from each feature plane of HexPlane could not guarantee high generation quality, mainly due to the absence of modeling spatial and temporal relations among the tokens.

**Padded Rollout Operation.** Given that the feature planes of HexPlane may share spatial or temporal dimensions, we employ the **P**added **R**ollout **O**peration (PRO) to systematically arrange all six planes into a unified square feature map, incorporating zero paddings in the uncovered corner areas. As shown in Fig. 4, the dimension of the 2D square feature map is $(\frac{X}{d_X} + \frac{Z}{d_Z} + \frac{T}{d_T})$, which minimizes the area for padding, where $d_X$, $d_Z$, and $d_T$ represent the downsampling rates along the X, Z, and T axes, respectively. Subsequently, we follow DiT to first "patchify" the constructed 2D feature map, converting it into a sequence of $N = ((\frac{X}{d_X} + \frac{Z}{d_Z} + \frac{T}{d_T})/p)^2$ tokens, where $p$ is the patch size, chosen so each token holds information from one feature plane. Following patchification, we apply the frequency-based positional embeddings to all tokens similar to DiT. Note that tokens corresponding to padding areas are excluded from the diffusion process. Consequently, the proposed PRO offers an efficient method for modeling spatial and temporal relationships within the token sequence.

**Conditional Generation.** DiT enables conditional generation through the use of Classifier-Free Guidance (CFG) (Ho & Salimans, 2022). To incorporate conditions into the generation process, we designed two branches for condition insertion (see Fig. 5). For any condition $c$, we use the adaLN-Zero technique from DiT, generating scale and shift parameters from $c$ and injecting them before and after the attention and feed-forward layers. To handle the complexity of image-based conditions, we add a cross-attention block to better integrate the image condition into the DiT block.

## 4.3 DOWNSTREAM APPLICATIONS

Beyond unconditional 4D scene generation, we explore novel applications of DynamicCity through conditional generation and HexPlane manipulation.

Table 1: **Comparisons of 4D Scene Reconstruction.** We report the mIoU scores of OccSora (Wang et al., 2024) and our DynamicCity framework on the *CarlaSC*, *Occ3D-Waymo*, and *Occ3D-nuScenes* datasets, respectively, under different resolutions and sequence lengths. Symbol $^\dagger$ denotes score reported in the OccSora paper. Other scores are reproduced using the official code.

| Dataset | #Classes | Resolution | #Frames | OccSora (Wang et al., 2024) | Ours (DynamicCity) |
|---|---|---|---|---|---|
| CarlaSC (Wilson et al., 2022) | 10 | 128×128×8 | 4 | 41.01% | **79.61%** (**+38.6%**) |
| | 10 | 128×128×8 | 8 | 39.91% | **76.18%** (**+36.3%**) |
| | 10 | 128×128×8 | 16 | 33.40% | **74.22%** (**+40.8%**) |
| | 10 | 128×128×8 | 32 | 28.91% | **59.31%** (**+30.4%**) |
| Occ3D-Waymo (Tian et al., 2023) | 9 | 200×200×16 | 16 | 36.38% | **68.18%** (**+31.8%**) |
| Occ3D-nuScenes (Tian et al., 2023) | 11 | 200×200×16 | 16 | 13.70% | **56.93%** (**+43.2%**) |
| | 11 | 200×200×16 | 32 | 13.51% | **42.60%** (**+29.1%**) |
| | 17 | 200×200×16 | 32 | 13.41% | **40.79%** (**+27.3%**) |
| | 17 | 200×200×16 | 32 | 27.40%$^\dagger$ | **40.79%** (**+13.4%**) |

Table 2: **Comparisons of 4D Scene Generation.** We report the Inception Score (IS), Fréchet Inception Distance (FID), Kernel Inception Distance (KID), and the Precision (P) and Recall (R) rates of OccSora (Wang et al., 2024) and our DynamicCity framework on the *CarlaSC* and *Occ3D-Waymo* datasets, respectively, in both the 2D and 3D spaces.

| Dataset | Method | #Frames | Metric[2D] | | | | | Metric[3D] | | | | |
|---|---|---|---|---|---|---|---|---|---|---|---|---|
| | | | IS[2D]↑ | FID[2D]↓ | KID[2D]↓ | P[2D]↑ | R[2D]↑ | IS[3D]↑ | FID[3D]↓ | KID[3D]↓ | P[3D]↑ | R[3D]↑ |
| CarlaSC (Wilson et al., 2022) | OccSora | 16 | 1.030 | 28.55 | 0.008 | 0.224 | 0.010 | 2.257 | 1559 | 52.72 | 0.380 | 0.151 |
| | Ours | | **1.040** | **12.94** | **0.002** | **0.307** | **0.018** | **2.331** | **354.2** | **19.10** | **0.460** | **0.170** |
| Occ3D-Waymo (Tian et al., 2023) | OccSora | 16 | 1.005 | 42.53 | 0.049 | 0.654 | 0.004 | 3.129 | 3140 | 12.20 | 0.384 | 0.001 |
| | Ours | | **1.010** | **36.73** | **0.001** | **0.705** | **0.015** | **3.206** | **1806** | 77.71 | **0.494** | **0.026** |

First, we showcase versatile uses of image conditions in the conditional generation pipeline: **1) HexPlane**: By autoregressively generating the HexPlane, we extend scene duration beyond temporal constraints. **2) Layout**: We control vehicle placement and dynamics in 4D scenes using conditions learned from bird's-eye view sketches.

To manage ego vehicle motion, we introduce two numerical conditioning methods: **3) Command**: Controls general ego vehicle motion via instructions. **4) Trajectory**: Enables fine-grained control through specific trajectory inputs.

Inspired by SemCity (Lee et al., 2024), we also manipulate the HexPlane during sampling to: **5) Inpaint**: Edit 4D scenes by masking HexPlane regions and guiding sampling with the masked areas. For more applications and implementation details, kindly refer to Sec. A.5 in the Appendix.

## 5 EXPERIMENTS

### 5.1 EXPERIMENTAL DETAILS

**Datasets.** We train the proposed model on the $^1$*Occ3D-Waymo*, $^2$*Occ3D-nuScenes*, and $^3$*CarlaSC* datasets. The former two from *Occ3D* (Tian et al., 2023) are derived from *Waymo* (Sun et al., 2020) and *nuScenes* (Caesar et al., 2020), where LiDAR point clouds have been completed and voxelized to form occupancy data. Each occupancy scene has a resolution of $200 \times 200 \times 16$, covering a region centered on the ego vehicle, extending 40 meters in all directions and 6.4 meters vertically. The *CarlaSC* dataset (Wilson et al., 2022) is a synthetic occupancy dataset, with a scene resolution of $128 \times 128 \times 8$, covering a region 25.6 meters around the ego vehicle, with a height of 3 meters.

**Implementation Details.** Our experiments are conducted using eight NVIDIA A100-80G GPUs. The global batch size used for training the VAE is 8, while the global batch size for training the DiT is 128. Our latent HexPlane $\mathcal{H}$ is compressed to half the size of the input $\mathbf{Q}$ in each dimension, with the latent channels $C = 16$. The weight for the Lovász-softmax and KL terms are set to 1 and 0.005, respectively. The learning rate for the VAE is $10^{-3}$, while the learning rate for the DiT is $10^{-4}$.

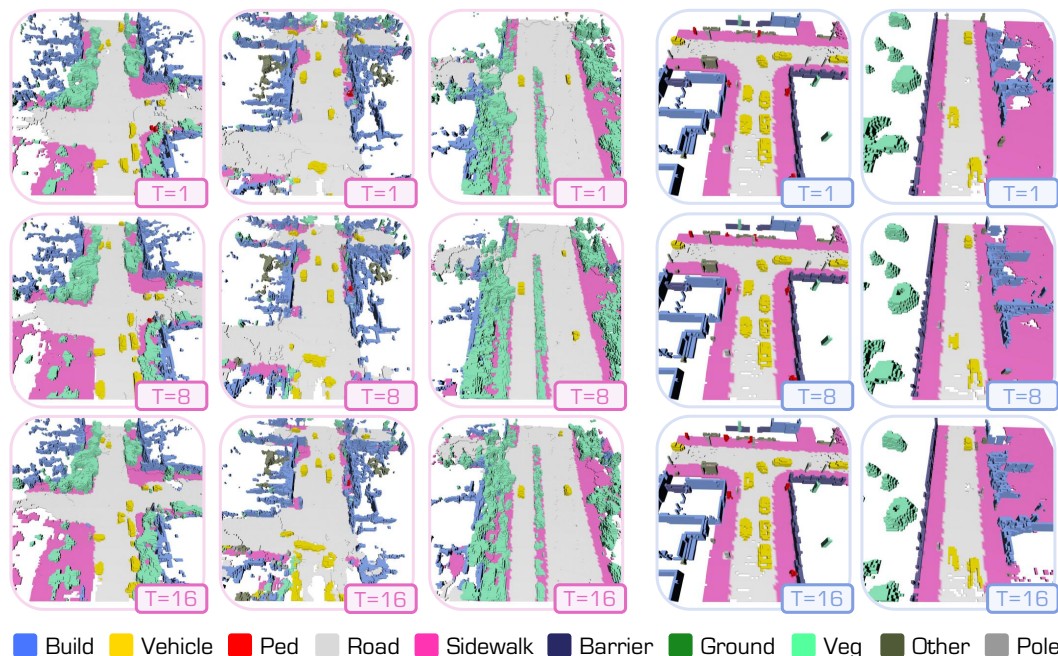

Build ■ Vehicle ■ Ped ■ Road ■ Sidewalk ■ Barrier ■ Ground ■ Veg ■ Other ■ Pole

Figure 6: **Dynamic Scene Generation Results.** We provide unconditional generation scenes from the 1st, 8th, and 16th frames on *Occ3D-Waymo* (Left) and *CarlaSC* (Right), respectively. Kindly refer to the Appendix for complete sequential scenes and longer temporal modeling examples.

**Evaluation Metrics.** The mean intersection over union (mIoU) metric is used to evaluate the reconstruction results of VAE. For DiT, Inception Score, FID, KID, Precision, and Recall are calculated for evaluation. Specifically, we follow prior work (Lee et al., 2024; Wang et al., 2024) by rendering 3D scenes into 2D images and utilizing conventional 2D evaluation pipelines for assessment. Additionally, we train the 3D Encoder to directly extract features from the 3D data and calculate the metrics. For more details, kindly refer to Sec. A.2 in the Appendix.

## 5.2 4D SCENE RECONSTRUCTION & GENERATION

**Reconstruction.** To evaluate the effectiveness of the proposed VAE in encoding the 4D occupancy sequence, we compare it with OccSora (Wang et al., 2024) using the CarlaSC, Occ3D-Waymo, and Occ3D-nuScenes datasets. As shown in Tab. 1, DynamicCity outperforms OccSora on these datasets, achieving mIoU improvements of 38.6%, 31.8%, and 43.2% respectively, when the input number of frames is 16. These results highlight the superior performance of the proposed VAE.

**Generation.** To demonstrate the effectiveness of DynamicCity in 4D scene generation, we compare the generation results with OccSora (Wang et al., 2024) on the Occ3D-Waymo and CarlaSC datasets. As shown in Tab. 2, the proposed method outperforms OccSora in terms of perceptual metrics in both 2D and 3D spaces. These results show that our model excels in both generation quality and diversity. Fig. 6 and Fig. 16 show the 4D scene generation results, demonstrating that our model is capable of generating large dynamic scenes in both real-world and synthetic datasets. Our model not only exhibits the ability to generate moving scenes with static semantics shifting as a whole, but it is also capable of generating dynamic elements such as vehicles and pedestrians.

**Applications.** Fig. 7 presents the results of our downstream applications. In tasks that involve inserting conditions into the DiT, such as command-conditional generation, trajectory-conditional generation, and layout-conditional generation, our model demonstrates the ability to generate reasonable scenes and dynamic elements while following the prompt to a certain extent. Additionally, the inpainting method proves that our HexPlane has explicit spatial meaning, enabling direct modifications within the scene by editing the HexPlane during inference.

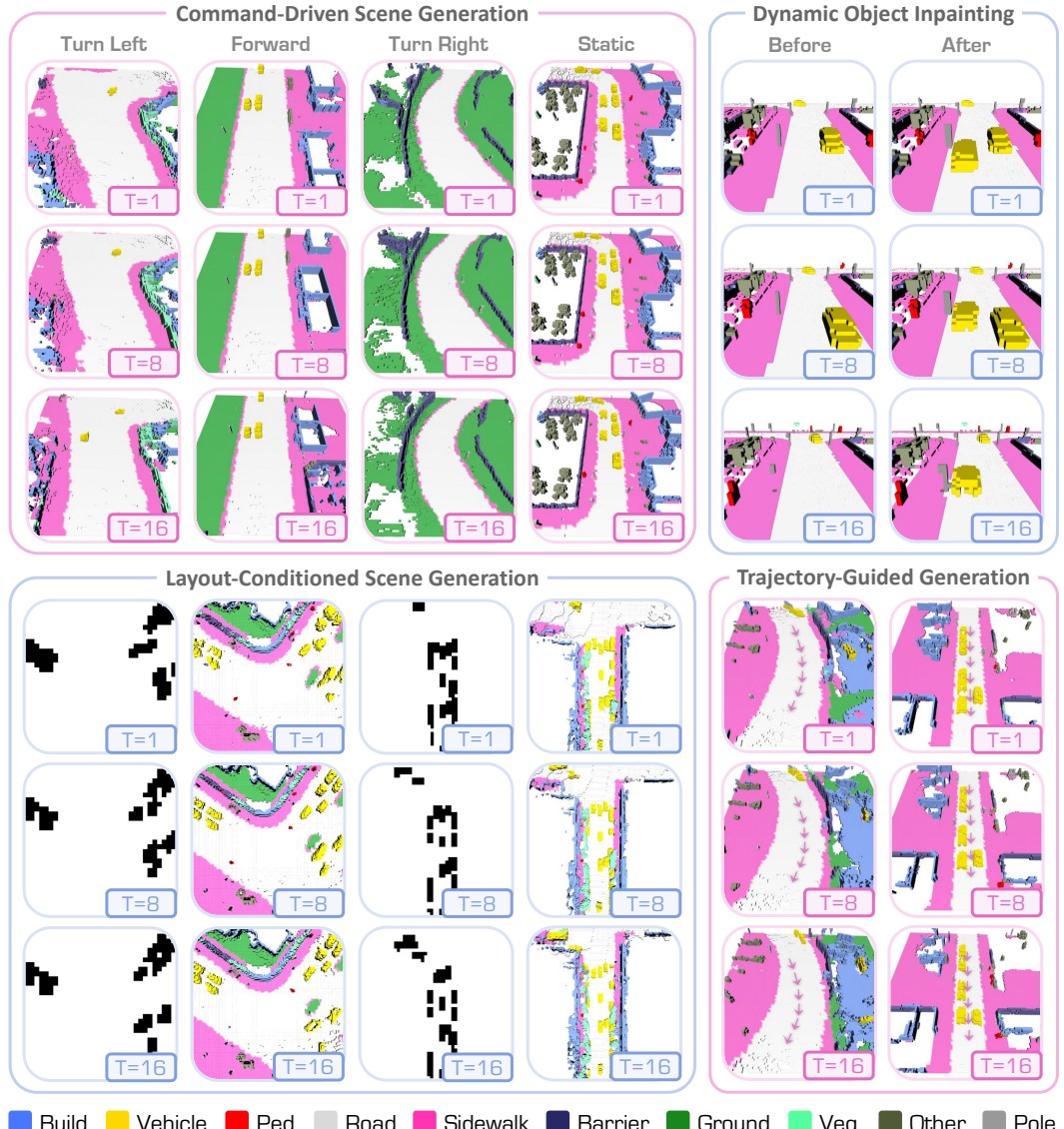

Figure 7: **Dynamic Scene Generation Applications.** We demonstrate the capability of our model on a diverse set of downstream tasks. We show the 1st, 8th, and 16th frames for simplicity. Kindly refer to the Appendix for complete sequential scenes and longer temporal modeling examples.

## 5.3 ABLATION STUDIES

We conduct ablation studies to demonstrate the effectiveness of the components of DynamicCity.

**VAE.** The effectiveness of the VAE is driven by two key innovations: Projection Module and Expansion & Squeeze Strategy (ESS). As shown in Tab. 3, the proposed Projection Module substantially improves HexPlane fitting performance, delivering up to a 12.56% increase in mIoU compared to traditional averaging operations. Additionally, compared to querying each point individually, ESS enhances HexPlane fitting quality with up to a 7.05% mIoU improvement, significantly boosts training speed by up to 2.06x, and reduces memory usage by a substantial 70.84%.

**HexPlane Dimensions.** The dimensions of HexPlane have a direct impact on both training efficiency and reconstruction quality. Table 4 provides a comparison of various downsample rates applied to the original HexPlane dimensions, which are $16 \times 128 \times 128 \times 8$ for CarlaSC and $16 \times 200 \times 200 \times 16$ for Occ3D-Waymo. As the downsampling rates increase, both the compression rate and training efficiency improve significantly, but the reconstruction quality, measured by mIoU, decreases.

Table 3: **Ablation Study on VAE Network Structures.** We report the mIoU scores, training time (second-per-iteration), and training-time memory consumption (VRAM) of different Encoder and Decoder configurations on *CarlaSC* and *Occ3D-Waymo*, respectively. Note that "ESS" denotes "Expansion & Squeeze". The **best** and second-best values are in **bold** and underlined.

| Encoder | Decoder | CarlaSC | | | Occ3D-Waymo | | |
|---|---|---|---|---|---|---|---|
| | | mIoU↑ | Time (s)↓ | VRAM (G)↓ | mIoU↑ | Time (s)↓ | VRAM (G)↓ |
| Average Pooling | Query | 60.97% | 0.236 | 12.46 | 49.37% | 1.563 | 69.66 |
| Average Pooling | ESS | 68.02% | **0.143** | **4.27** | 55.72% | **0.758** | **20.31** |
| Projection | Query | 68.73% | 0.292 | 13.59 | 61.93% | 2.128 | 73.15 |
| Projection | ESS | **74.22%** | 0.205 | 5.92 | **62.57%** | 1.316 | 25.92 |

Table 4: **Ablation Study on HexPlane Downsampling (D.S.) Rates.** We report the compression ratios (C.R.), mIoU scores, training speed (seconds per iteration), and training-time memory consumption on *CarlaSC* and *Occ3D-Waymo*. The **best** and second-best values are in **bold** and underlined.

| D.S. Rates | | | | CarlaSC | | | | Occ3D-Waymo | | | |
|---|---|---|---|---|---|---|---|---|---|---|---|
| $d_T$ | $d_X$ | $d_Y$ | $d_Z$ | C.R.↑ | mIoU↑ | Time (s)↓ | VRAM (G)↓ | C.R.↑ | mIoU↑ | Time (s)↓ | VRAM (G)↓ |
| 1 | 1 | 1 | 1 | 5.78% | **84.67%** | 1.149 | 21.63 | Out-of-Memory | | | >80 |
| 1 | 2 | 2 | 1 | 17.96% | 76.05% | 0.289 | 8.49 | 38.42% | **63.30%** | 1.852 | 32.82 |
| 2 | 2 | 2 | 2 | 23.14% | 74.22% | 0.205 | 5.92 | 48.25% | 62.37% | 0.935 | 24.9 |
| 2 | 4 | 4 | 2 | **71.86%** | 65.15% | **0.199** | **4.00** | **153.69%** | 58.13% | **0.877** | **22.30** |

Table 5: **Ablation Study on Organizing HexPlane as Image Tokens.** We report the Inception Score (IS), Fréchet Inception Distance (FID), Kernel Inception Distance (KID), and the Precision (P) and Recall (R) rates on *CarlaSC*. The **best** values are highlighted in **bold**.

| Method | Metric[2D] | | | | | Metric[3D] | | | | |
|---|---|---|---|---|---|---|---|---|---|---|
| | IS[2D]↑ | FID[2D]↓ | KID[2D]↓ | P[2D]↑ | R[2D]↑ | IS[3D]↑ | FID[3D]↓ | KID[3D]↓ | P[3D]↑ | R[3D]↑ |
| Direct Unfold | 2.496 | 205.0 | 0.248 | 0.000 | 0.000 | 2.269 | 9110 | 723.7 | 0.173 | 0.043 |
| Vertical Concatenation | 2.476 | 12.79 | 0.003 | 0.191 | 0.042 | 2.305 | 623.2 | 26.67 | 0.424 | 0.159 |
| Padded Rollout | **2.498** | **10.96** | **0.002** | **0.238** | **0.066** | **2.331** | **354.2** | **19.10** | **0.460** | **0.170** |

To achieve the optimal balance between training efficiency and reconstruction quality, we select a downsampling rate of $d_T = d_X = d_Y = d_Z = 2$.

**Padded Rollout Operation.** We compare the Padded Rollout Operation with different strategies for obtaining image tokens: **1)** Direct Unfold: directly unfolding the six planes into patches and concatenating them; **2)** Vertical Concat: vertically concatenating the six planes without aligning dimensions during the rollout process. As shown in Tab. 5, Padded Rollout Operation (PRO) efficiently models spatial and temporal relationships in the token sequence, achieving optimal generation quality.

## 6 CONCLUSION

We present DynamicCity, a framework for high-quality 4D occupancy generation that captures the temporal dynamics of real-world environments. Our method introduces HexPlane, a compact 4D representation generated using a VAE with a Projection Module, alongside an Expansion & Squeeze Strategy to enhance reconstruction efficiency and accuracy. Additionally, our Masked Rollout Operation reorganizes HexPlane features for DiT-based diffusion, enabling versatile 4D scene generation. Extensive experiments demonstrate that DynamicCity surpasses state-of-the-art methods in both reconstruction and generation, offering significant improvements in quality, training speed, and memory efficiency. DynamicCity paves the way for future research in dynamic scene generation.

## ACKNOWLEDGMENTS

This study is supported by the National Key R&D Program of China No.2022ZD0160102. This study is also supported by the Ministry of Education, Singapore, under its MOE AcRF Tier 2 (MOE-T2EP20221-0012, MOE-T2EP20223-0002), NTU NAP, and under the RIE2020 Industry Alignment Fund – Industry Collaboration Projects (IAF-ICP) Funding Initiative, as well as cash and in-kind contribution from the industry partner(s). This study is also under the programme DesCartes and is supported by the National Research Foundation, Prime Minister's Office, Singapore, under its Campus for Research Excellence and Technological Enterprise (CREATE) programme.

We sincerely thank our reviewers and ACs for the time and effort they devoted during the review.

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

APPENDIX

In this appendix, we supplement the following materials to support the findings and conclusions drawn in the main body of this paper.

# A    ADDITIONAL IMPLEMENTATION DETAILS

In this section, we provide additional implementation details to assist in reproducing this work. Specifically, we elaborate on the details of the datasets, DiT evaluation metrics, the specifics of our generation models, and discussions on the downstream applications.

## A.1    DATASETS

Our experiments primarily utilize two datasets: *Occ3D-Waymo* (Tian et al., 2023) and *CarlaSC* (Wilson et al., 2022). Additionally, we also evaluate our VAE on *Occ3D-nuScenes* (Tian et al., 2023).

The *Occ3D-Waymo* dataset is derived from real-world Waymo Open Dataset (Sun et al., 2020) data, where occupancy sequences are obtained through multi-frame fusion and voxelization processes. Similarly, *Occ3D-nuScenes* is generated from the real-world nuScenes (Caesar et al., 2020) dataset using the same fusion and voxelization operations. On the other hand, the *CarlaSC* dataset is generated from simulated scenes and sensor data, yielding occupancy sequences.

Using these different datasets demonstrates the effectiveness of our method on both real-world and synthetic data. To ensure consistency in the experimental setup, we select 11 commonly used semantic categories and map the original categories from both datasets to these 11 categories. The detailed semantic label mappings are provided in Tab. 6.

Table 6: **Summary of Semantic Label Mappings.** We unify the semantic classes between *CarlaSC* (Wilson et al., 2022), *Occ3D-Waymo* (Tian et al., 2023), and *Occ3D-nuScenes* (Tian et al., 2023) datasets for semantic scene generation.

| Class | CarlaSC | Occ3D-Waymo | Occ3D-nuScenes |
|---|---|---|---|
| ■ Building | Building | Building | Manmade |
| ■ Barrier | Barrier, Wall, Guardrail | - | Barrier |
| ■ Other | Other, Sky, Bridge, Rail track, Static, Dynamic, Water | General Object | General Object |
| ■ Pedestrian | Pedestrian | Pedestrian | Pedestrian |
| ■ Pole | Pole, Traffic sign, Traffic light | Sign, Traffic light, Pole, Construction Cone | Traffic cone |
| ■ Road | Road, Roadlines | Road | Drivable surface |
| ■ Ground | Ground, Terrain | - | Other flat, Terrain |
| ■ Sidewalk | Sidewalk | Sidewalk | Sidewalk |
| ■ Vegetation | Vegetation | Vegetation, Tree trunk | Vegetation |
| ■ Vehicle | Vehicle | Vehicle | Bus, Car, Construction vehicle, Trailer, Truck |
| ■ Bicycle | - | Bicyclist, Bicycle, Motorcycle | Bicycle, Motorcycle |

- **Occ3D-Waymo.** This dataset contains 798 training scenes, with each scene lasting approximately 20 seconds and sampled at a frequency of 10 Hz. This dataset includes 15 semantic categories. We use volumes with a resolution of $200 \times 200 \times 16$ from this dataset.
- **CarlaSC.** This dataset contains 6 training scenes, each duplicated into Light, Medium, and Heavy based on traffic density. Each scene lasts approximately 180 seconds and is sampled at a frequency of 10 Hz. This dataset contains 22 semantic categories, and the scene resolution is $128 \times 128 \times 8$.
- **Occ3D-nuScenes.** This dataset contains 600 scenes, with each scene lasting approximately 20 seconds and sampled at a frequency of 2 Hz. Compared to Occ3D-Waymo and CarlaSC, Occ3D-nuScenes has fewer total frames and more variation between scenes. This dataset includes 17 semantic categories, with a resolution of $200 \times 200 \times 16$.

## A.2 DiT Evaluation Metrics

**Inception Score (IS).** This metric evaluates the quality and diversity of generated samples using a pre-trained Inception model as follows:

$$\text{IS} = \exp\left(\mathbb{E}_{\mathbf{Q}\sim p_g}\left[D_{\text{KL}}(p(y|\mathbf{Q}) \parallel p(y))\right]\right) , \tag{5}$$

where $p_g$ represents the distribution of generated samples. $p(y|\mathbf{Q})$ is the conditional label distribution given by the Inception model for a generated sample $\mathbf{Q}$. $p(y) = \int p(y|\mathbf{Q})p_g(\mathbf{Q}) \, d\mathbf{Q}$ is the marginal distribution over all generated samples. $D_{\text{KL}}(p(y|\mathbf{Q}) \parallel p(y))$ is the Kullback-Leibler divergence, defined as follows:

$$D_{\text{KL}}(p(y|\mathbf{Q}) \parallel p(y)) = \sum_i p(y_i|\mathbf{Q}) \log \frac{p(y_i|\mathbf{Q})}{p(y_i)} . \tag{6}$$

**Fréchet Inception Distance (FID).** This metric measures the distance between the feature distributions of real and generated samples:

$$\text{FID} = \|\mu_r - \mu_g\|^2 + \text{Tr}\left(\Sigma_r + \Sigma_g - 2(\Sigma_r\Sigma_g)^{1/2}\right) , \tag{7}$$

where $\mu_r$ and $\Sigma_r$ are the mean and covariance matrix of features from real samples. $\mu_g$ and $\Sigma_g$ are the mean and covariance matrix of features from generated samples. $\text{Tr}$ denotes the trace of a matrix.

**Kernel Inception Distance (KID).** This metric uses the squared Maximum Mean Discrepancy (MMD) with a polynomial kernel as follows:

$$\text{KID} = \text{MMD}^2(\phi(\mathbf{Q}_r), \phi(\mathbf{Q}_g)) , \tag{8}$$

where $\phi(\mathbf{Q}_r)$ and $\phi(\mathbf{Q}_g)$ represent the features of real and generated samples extracted from the Inception model.

MMD with a polynomial kernel $k(x, y) = (x^\top y + c)^d$ is calculated as follows:

$$\text{MMD}^2(X, Y) = \frac{1}{m(m-1)} \sum_{i\neq j} k(x_i, x_j) + \frac{1}{n(n-1)} \sum_{i\neq j} k(y_i, y_j) - \frac{2}{mn} \sum_{i,j} k(x_i, y_j) , \tag{9}$$

where $X = \{\mathbf{Q}_1, \ldots, \mathbf{Q}_m\}$ and $Y = \{\mathbf{y}_1, \ldots, \mathbf{y}_n\}$ are sets of features from real and generated samples.

**Precision.** This metric measures the fraction of generated samples that lie within the real data distribution as follows:

$$\text{Precision} = \frac{1}{N} \sum_{i=1}^{N} \mathbb{I}\left((\mathbf{f}_g - \mu_r)^\top \Sigma_r^{-1}(\mathbf{f}_g - \mu_r) \leq \chi^2\right) , \tag{10}$$

where $\mathbf{f}_g$ is a generated sample in the feature space. $\mu_r$ and $\Sigma_r$ are the mean and covariance of the real data distribution. $\mathbb{I}(\cdot)$ is the indicator function. $\chi^2$ is a threshold based on the chi-squared distribution.

**Recall.** This metric measures the fraction of real samples that lie within the generated data distribution as follows:

$$\text{Recall} = \frac{1}{M} \sum_{j=1}^{M} \mathbb{I}\left((\mathbf{f}_r - \mu_g)^\top \Sigma_g^{-1}(\mathbf{f}_r - \mu_g) \leq \chi^2\right) , \tag{11}$$

where: $\mathbf{f}_r$ is a real sample in the feature space. $\mu_g$ and $\Sigma_g$ are the mean and covariance of the generated data distribution. $\mathbb{I}(\cdot)$ is the indicator function. $\chi^2$ is a threshold based on the chi-squared distribution.

**2D Evaluations.** We render 3D scenes as 2D images for 2D evaluations. To ensure fair comparisons, we use the same semantic colormap and camera settings across all experiments. Fig. 8 shows an example of a rendered 2D semantic colormap. We use an InceptionV3 (Szegedy et al., 2015) model to compute the Inception Score (IS), Fréchet Inception Distance (FID), and Kernel Inception Distance (KID) scores, while Precision and Recall are computed using a VGG-16 (Simonyan & Zisserman, 2015) model. We train both 2D backbones using semantic colormap data.

**3D Evaluations.** For 3D data, we train a MinkowskiUNet (Choy et al., 2019) as an autoencoder. We adopt the latest implementation from SPVNAS (Tang et al., 2020), which supports optimized sparse convolution operations. The features were extracted by applying average pooling to the output of the final downsampling block.

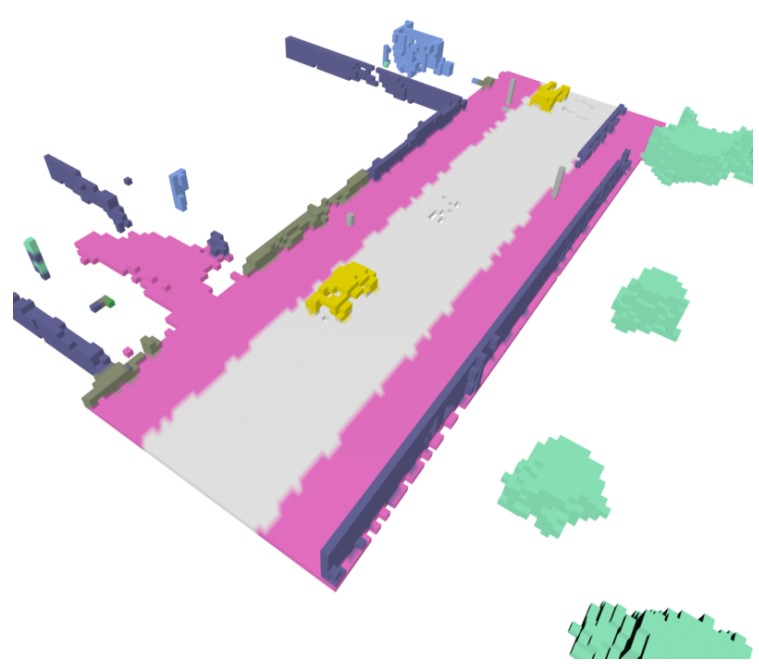

Figure 8: **Example of 2D Evaluation Rendering.**

### A.3  MODEL DETAILS

**General Training Details.** We implement both the VAE and DiT models using PyTorch (Paszke et al., 2019). We utilize PyTorch's mixed precision and replace all attention mechanisms with FlashAttention (Dao et al., 2022) to accelerate training and reduce memory usage. AdamW is used as the optimizer for all models.

We train the VAE with a learning rate of $10^{-3}$, running for 20 epochs on Occ3D-Waymo and 100 epochs on CarlaSC. The DiT is trained with a learning rate of $10^{-4}$, and the EMA rate for DiT is set to 0.9999.

**VAE.** Our encoder projects the 4D input $\mathbf{Q}$ into a HexPlane, where each dimension is a compressed version of the original 4D input. First, a 3D CNN is applied to each frame for feature extraction and downsampling, with dimensionality reduction applied only to the spatial dimensions $(X, Y, Z)$. Next, the Projection Module projects the 4D features into the HexPlane. Each small transformer within the Projection Module consists of two layers, and the attention mechanism has two heads. Each head has a dimensionality of 16, with a dropout rate of 0.1. Afterward, we further downsample the $T$ dimension to half of its original size.

During decoding, we first use three small transpose CNNs to restore the $T$ dimension, then use an ESS module to restore the 4D features. Finally, we apply a 3D CNN to recover the spatial dimensions and generate point-wise predictions.

**Diffusion.** We set the patch size $p$ to 2 for our DiT models. The Waymo DiT model has a hidden size of 768, 18 DiT blocks, and 12 attention heads. The CarlaSC DiT model has a hidden size of 384, 16 DiT blocks, and 8 attention heads.

**Discussion on VAE Structure Improvements.** Some prior work utilizes sparse 3D structures to enhance the efficiency of their backbones. For example, XCube (Ren et al., 2024b) employs a fully sparse 3D encoder, significantly improving model efficiency. Similarly, our VAE could potentially improve the 3D convolutional feature extractor $\boldsymbol{f}_\theta(\cdot)$ by adopting sparse convolution. However, using sparse convolution offers only limited efficiency gains, as convolution accounts for only a small

portion of our VAE. Moreover, like XCube, we cannot apply sparse convolution in our decoder. In the future, we plan to explore more efficient operations to further optimize our 3D backbone.

### A.4 CLASSIFIER-FREE GUIDANCE

Classifier-Free Guidance (CFG) (Ho & Salimans, 2022) could improve the performance of conditional generative models without relying on an external classifier. Specifically, during training, the model simultaneously learns both conditional generation $p(x|c)$ and unconditional generation $p(x)$, and guidance during sampling is provided by the following equation:

$$\hat{x}_t = (1 + w) \cdot \hat{x}_t(c) - w \cdot \hat{x}_t(\emptyset) , \tag{12}$$

where $\hat{x}_t(c)$ is the result conditioned on $c$, $\hat{x}_t(\emptyset)$ is the unconditioned result, and $w$ is a weight parameter controlling the strength of the conditional guidance. By adjusting $w$, an appropriate balance between the accuracy and diversity of the generated scenes can be achieved.

### A.5 DOWNSTREAM APPLICATIONS

This section provides a comprehensive explanation of five tasks to demonstrate the capability of our 4D scene generation model across various scenarios.

**HexPlane.** Since our model is based on Latent Diffusion Models, it is inherently constrained to generate results that match the latent space dimensions, limiting the temporal length of unconditionally generated sequences. We argue that a robust 4D generation model should not be restricted to producing only short sequences. Instead of increasing latent space size, we leverage CFG to generate sequences in an auto-regressive manner. By conditioning each new 4D sequence on the previous one, we sequentially extend the temporal dimension. This iterative process significantly extends sequence length, enabling long-term generation, and allows conditioning on any real-world 4D scene to predict the next sequence using the DiT model. Theoretically, our HexPlane conditional generation can model sequence of arbitrary length, but less stable generation may occur when generating very long sequences.

We condition our DiT by using the HexPlane from $T$ frames earlier. For any condition HexPlane, we apply patch embedding and positional encoding operations to obtain condition tokens. These tokens, combined with other conditions, are fed into the adaLN-Zero and Cross-Attention branches to influence the main branch.

**Layout.** To control object placement in the scene, we train a model capable of generating vehicle dynamics based on a bird's-eye view sketch. We apply semantic filtering to the bird's-eye view of the input scene, marking regions with vehicles as $1$ and regions without vehicles as $0$. Pooling this binary image provides layout information as a $T \times H \times W$ tensor from the bird's-eye perspective. The layout is padded to match the size of the HexPlane, ensuring that the positional encoding of the bird's-eye layout aligns with the $XY$ plane. DiT learns the correspondence between the layout and vehicle semantics using the same conditional injection method applied to the HexPlane.

**Command.** While we have developed effective methods to control the HexPlane in both temporal and spatial dimensions, a critical aspect of 4D autonomous driving scenarios is the motion of the ego vehicle. To address this, we define four commands: `STATIC`, `FORWARD`, `TURN LEFT`, and `TURN RIGHT`, and annotate our training data by analyzing ego vehicle poses. During training, we follow the traditional DiT approach of injecting class labels, where the commands are embedded and fed into the model via adaLN-Zero.

**Trajectory.** For more fine-grained control of the ego vehicle's motion, we extend the command-based conditioning into a trajectory condition branch. For any 4D scene, the $XY$ coordinates of the trajectory $\text{traj} \in \mathbb{R}^{T \times 2}$ are passed through an MLP and injected into the adaLN-Zero branch.

**Inpaint.** We demonstrate that our model can handle versatile applications by training a conditional DiT for the previous tasks. Extending our exploration of downstream applications, and inspired by (Lee et al., 2024), we leverage the 2D structure of our latent space and the explicit modeling of each dimension to highlight our model's ability to perform inpainting on 4D scenes. During DiT sampling, we define a 2D mask $m \in \mathbb{R}^{X \times Y}$ on the $XY$ plane, which is extended across all dimensions to mask specific regions of the HexPlane.

At each step of the diffusion process, we apply noise to the input $\mathcal{H}^{\text{in}}$ and update the HexPlane using the following formula:

$$\mathcal{H}_t = m \odot \mathcal{H}_t + (1 - m) \odot \mathcal{H}_t^{\text{in}} \, , \tag{13}$$

where $\odot$ denotes the element-wise product. This process inpaints the masked regions while preserving the unmasked areas of the scene, enabling partial scene modification, such as turning an empty street into one with heavy traffic.

**Outpaint.** Outpainting extends the spatial dimensions of a given occupancy sequence. We use the same procedure for outpainting as we do for inpainting. Specifically, we mask half of the scene, shift the latent representation, and apply the inpainting process. Consequently, we could obtain a larger scene with consistent dynamics.

**Single frame occupancy.** We apply the same procedure for single-frame occupancy conditional generation as for HexPlane conditional generation. Specifically, we preprocess the data, encode the first frame of each training sequence as a HexPlane, and fine-tune our HexPlane generation model for single-frame conditional generation.

# B ADDITIONAL QUANTITATIVE RESULTS

In this section, we present additional quantitative results to demonstrate the effectiveness of our VAE in accurately reconstructing 4D scenes.

## B.1 PER-CLASS GENERATION RESULTS

We include the class-wise IoU scores of OccSora (Wang et al., 2024) and our proposed DynamicCity framework on *CarlaSC* (Wilson et al., 2022). As shown in Tab. 7, our results demonstrate higher IoU across all classes, indicating that our VAE reconstruction achieves minimal information loss. Additionally, our model does not exhibit significantly low IoU for any specific class, proving its ability to effectively handle class imbalance.

Table 7: **Comparisons of Per-Class IoU Scores.** We compared the performance of OccSora (Wang et al., 2024), and our DynamicCity framework on *CarlaSC* (Wilson et al., 2022) across 10 semantic classes. The scene resolution is $128 \times 128 \times 8$. The sequence lengths are 4, 8, 16, and 32, respectively.

| Method | mIoU | Building | Barrier | Other | Pedestrian | Pole | Road | Ground | Sidewalk | Vegetation | Vehicle |
|---|---|---|---|---|---|---|---|---|---|---|---|
| **Resolution:** $128 \times 128 \times 8$ | | **Sequence Length:** 4 | | | | | | | | | |
| OccSora | 41.009 | 38.861 | 10.616 | 6.637 | 19.191 | 21.825 | 93.910 | 61.357 | 86.671 | 15.685 | 55.340 |
| **Ours** | 79.604 | 76.364 | 31.354 | 68.898 | 93.436 | 87.962 | 98.617 | 87.014 | 95.129 | 68.700 | 88.569 |
| *Improv.* | 38.595 | 37.503 | 20.738 | 62.261 | 74.245 | 66.137 | 4.707 | 25.657 | 8.458 | 53.015 | 33.229 |
| **Resolution:** $128 \times 128 \times 8$ | | **Sequence Length:** 8 | | | | | | | | | |
| OccSora | 39.910 | 33.001 | 3.260 | 5.659 | 19.224 | 19.357 | 93.038 | 57.335 | 85.551 | 30.899 | 51.776 |
| **Ours** | 76.181 | 70.874 | 50.025 | 52.433 | 87.958 | 85.866 | 97.513 | 83.074 | 93.944 | 58.626 | 81.498 |
| *Improv.* | 36.271 | 37.873 | 46.765 | 46.774 | 68.734 | 66.509 | 4.475 | 25.739 | 8.393 | 27.727 | 29.722 |
| **Resolution:** $128 \times 128 \times 8$ | | **Sequence Length:** 16 | | | | | | | | | |
| OccSora | 33.404 | 19.264 | 2.205 | 3.454 | 11.781 | 9.165 | 92.054 | 50.077 | 82.594 | 18.078 | 45.363 |
| **Ours** | 74.223 | 66.852 | 51.901 | 49.844 | 79.410 | 82.369 | 96.937 | 84.484 | 94.082 | 58.217 | 78.134 |
| *Improv.* | 40.819 | 47.588 | 49.696 | 46.390 | 67.629 | 73.204 | 4.883 | 34.407 | 11.488 | 40.139 | 32.771 |
| **Resolution:** $128 \times 128 \times 8$ | | **Sequence Length:** 32 | | | | | | | | | |
| OccSora | 28.911 | 16.565 | 1.413 | 0.944 | 6.200 | 4.150 | 91.466 | 43.399 | 78.614 | 11.007 | 35.353 |
| **Ours** | 59.308 | 52.036 | 25.521 | 29.382 | 56.811 | 57.876 | 94.792 | 78.390 | 89.955 | 46.080 | 62.234 |
| *Improv.* | 30.397 | 35.471 | 24.108 | 28.438 | 50.611 | 53.726 | 3.326 | 34.991 | 11.341 | 35.073 | 26.881 |

## B.2 OCCUPANCY FORECASTING RESULTS

We train our HexPlane conditional generation pipeline on Occ3D-nuScenes (Tian et al., 2023) as an occupancy forecasting model. We set $T = 4$ to ensure the model receives a HexPlane with a context length of 2 seconds, aligning with OccWorld (Zheng et al., 2024a), and generates the next 2 seconds for evaluation. As shown in Tab. 8, our model outperforms OccWorld on most metrics.

Table 8: **4D Occupancy Forecasting Performance**. We compare the performance of OccWorld (Zheng et al., 2024a) and our proposed DynamicCity framework on Occ3D-nuScenes (Tian et al., 2023).

| Method | mIoU | | | IoU | | |
|---|---|---|---|---|---|---|
| | T = 0 | T = 1 | T = 2 | T = 0 | T = 1 | T = 2 |
| OccWorld-O | 66.38 | 25.78 | 15.14 | 62.29 | **34.63** | 25.07 |
| **Ours** | **80.52** | **26.18** | **16.94** | **67.64** | 34.12 | **25.82** |

### B.3  USER STUDY

We conduct a user study comparing OccSora (Wang et al., 2024) with our proposed DynamicCity. The study includes 20 samples, with 10 from each method. Participants rate each sample on four metrics: 1) overall quality, 2) time consistency, 3) background quality, and 4) foreground quality. Ratings range from 1 to 5, with 5 being the highest. We collect results from 42 volunteers and get 840 valid scores in total, as shown in Tab. 9. Our method receives better user feedback across all metrics.

Table 9: **User Study Results.** We conduct user study comparing OccSora (Wang et al., 2024) and DynamicCity. The rating is of scale 1-5, the higher the better.

| Method | Overall Quality | Time Consistency | Background Quality | Foreground Quality |
|--------|-----------------|------------------|--------------------|--------------------|
| OccSora | 2.21 | 2.05 | 2.17 | 2.11 |
| **Ours** | **4.03** | **4.02** | **3.95** | **4.04** |

### B.4  MODEL STATS

We compare the training speed, inference speed, training VRAM, and inference VRAM of Occ-Sora (Wang et al., 2024) and DynamicCity. The results are presented in Tab. 10, Tab. 11, Tab. 12, and Tab. 13. While some of our models may be slightly slower and consume more memory compared to OccSora, they achieve significantly better performance. We also compare the total model size of OccSora and our model in Tab. 14. Our model is significantly smaller than OccSora while achieving superior performance.

Table 10: **VAE Model Statistics on CarlaSC Dataset (Wilson et al., 2022).** We compare the training time, inference time, training VRAM, inference VRAM of OccSora (Wang et al., 2024) and our DynamicCity.

| Method | Training Time (s)↓ | Inference Time (s)↓ | Training VRAM (G)↓ | Inference VRAM (G)↓ |
|--------|--------------------|---------------------|--------------------|---------------------|
| OccSora | 0.36 | 0.21 | 4.86 | 3.25 |
| **Ours** | 0.21 | 0.41 | 5.92 | 1.43 |

Table 11: **VAE Model Statistics on Occ3D-Waymo Dataset (Tian et al., 2023).** We compare the training time, inference time, training VRAM, inference VRAM of OccSora (Wang et al., 2024) and our DynamicCity.

| Method | Training Time (s)↓ | Inference Time (s)↓ | Training VRAM (G)↓ | Inference VRAM (G)↓ |
|--------|--------------------|---------------------|--------------------|---------------------|
| OccSora | 0.63 | 0.21 | 10.05 | 3.93 |
| **Ours** | 0.94 | 0.54 | 24.90 | 4.62 |

### B.5  COMPARISONS WITH SEMCITY

We compare the generation quality of SemCity (Lee et al., 2024) and our DynamicCity in Tab. 15. Despite using a more compact latent representation and generating dynamic scenes, our model outperforms SemCity on most metrics.

Table 12: **DiT Model Statistics on CarlaSC Dataset** (Wilson et al., 2022). We compare the training time, inference time, training VRAM, inference VRAM of OccSora (Wang et al., 2024) and our DynamicCity.

| Method | Training Time (s)↓ | Inference Time (s)↓ | Training VRAM (G)↓ | Inference VRAM (G)↓ |
|---|---|---|---|---|
| OccSora | 0.19 | 6.10 | 1.50 | 1.15 |
| **Ours** | 0.19 | 3.91 | 10.22 | 1.28 |

Table 13: **DiT Model Statistics on Occ3D-Waymo Dataset** (Tian et al., 2023). We compare the training time, inference time, training VRAM, inference VRAM of OccSora (Wang et al., 2024) and our DynamicCity.

| Method | Training Time (s)↓ | Inference Time (s)↓ | Training VRAM (G)↓ | Inference VRAM (G)↓ |
|---|---|---|---|---|
| OccSora | 0.35 | 6.09 | 15.16 | 1.15 |
| **Ours** | 0.45 | 4.41 | 22.33 | 1.29 |

Table 14: **Model Size.** We compare the total model size of OccSora (Wang et al., 2024) and our DynamicCity.

| Dataset | Method | Model Size (M) |
|---|---|---|
| CarlaSC | OccSora | 169.1 |
| | **Ours** | 44.7 |
| Occ3D-Waymo | OccSora | 174.2 |
| | **Ours** | 45.6 |

Table 15: **Comparisons of 2D and 3D Evaluation Metrics.** We report the Inception Score (IS), Fréchet Inception Distance (FID), Kernel Inception Distance (KID), and the Precision (P) and Recall (R) rates for SemCity (Lee et al., 2024) and our method in both the **2D** and **3D** spaces.

| Method | Metric[2D] | | | | | Metric[3D] | | | | |
|---|---|---|---|---|---|---|---|---|---|---|
| | IS↑ | FID↓ | KID↓ | P↑ | R↑ | IS↑ | FID↓ | KID↓ | P↑ | R↑ |
| SemCity | 1.039 | 35.40 | 0.010 | 0.213 | **0.058** | 2.288 | 1113 | 53.948 | 0.253 | **0.787** |
| **Ours** | **1.040** | **12.94** | **0.002** | **0.307** | 0.018 | **2.331** | 427.5 | 27.869 | **0.460** | 0.170 |

## C    ADDITIONAL QUALITATIVE RESULTS

In this section, we provide additional qualitative results on the *Occ3D-Waymo* (Tian et al., 2023) and *CarlaSC* (Wilson et al., 2022) datasets to demonstrate the effectiveness of our approach.

### C.1    UNCONDITIONAL DYNAMIC SCENE GENERATION

First, we present full unconditional generation results in Fig. 9 and 10. These results demonstrate that our generated scenes are of high quality, realistic, and contain significant detail, capturing both the overall scene dynamics and the movement of objects within the scenes.

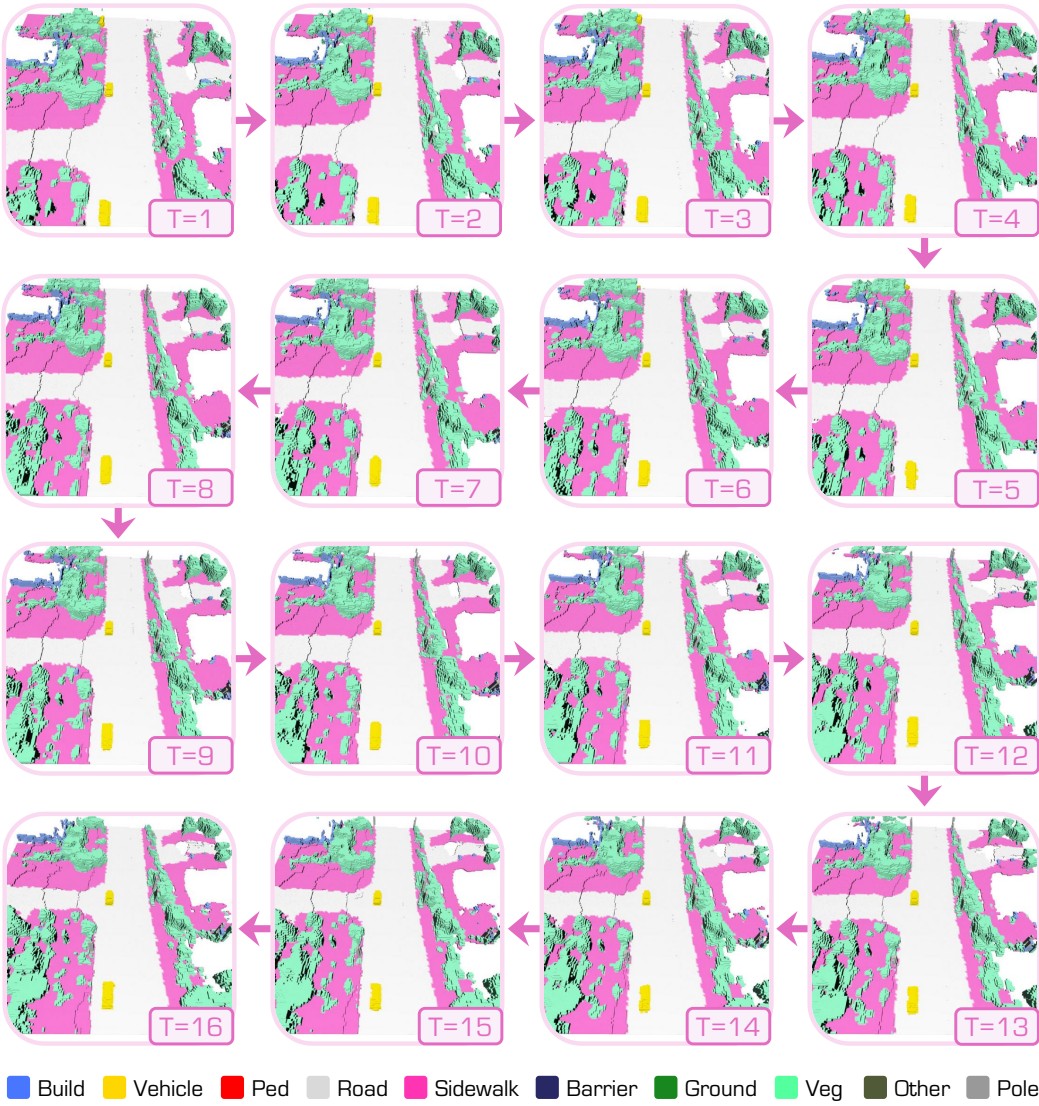

Figure 9: **Unconditional Dynamic Scene Generation Results.** We provide qualitative examples of a total of 16 consecutive frames generated by DynamicCity on the *Occ3D-Waymo* (Tian et al., 2023) dataset. Best viewed in colors and zoomed-in for additional details.

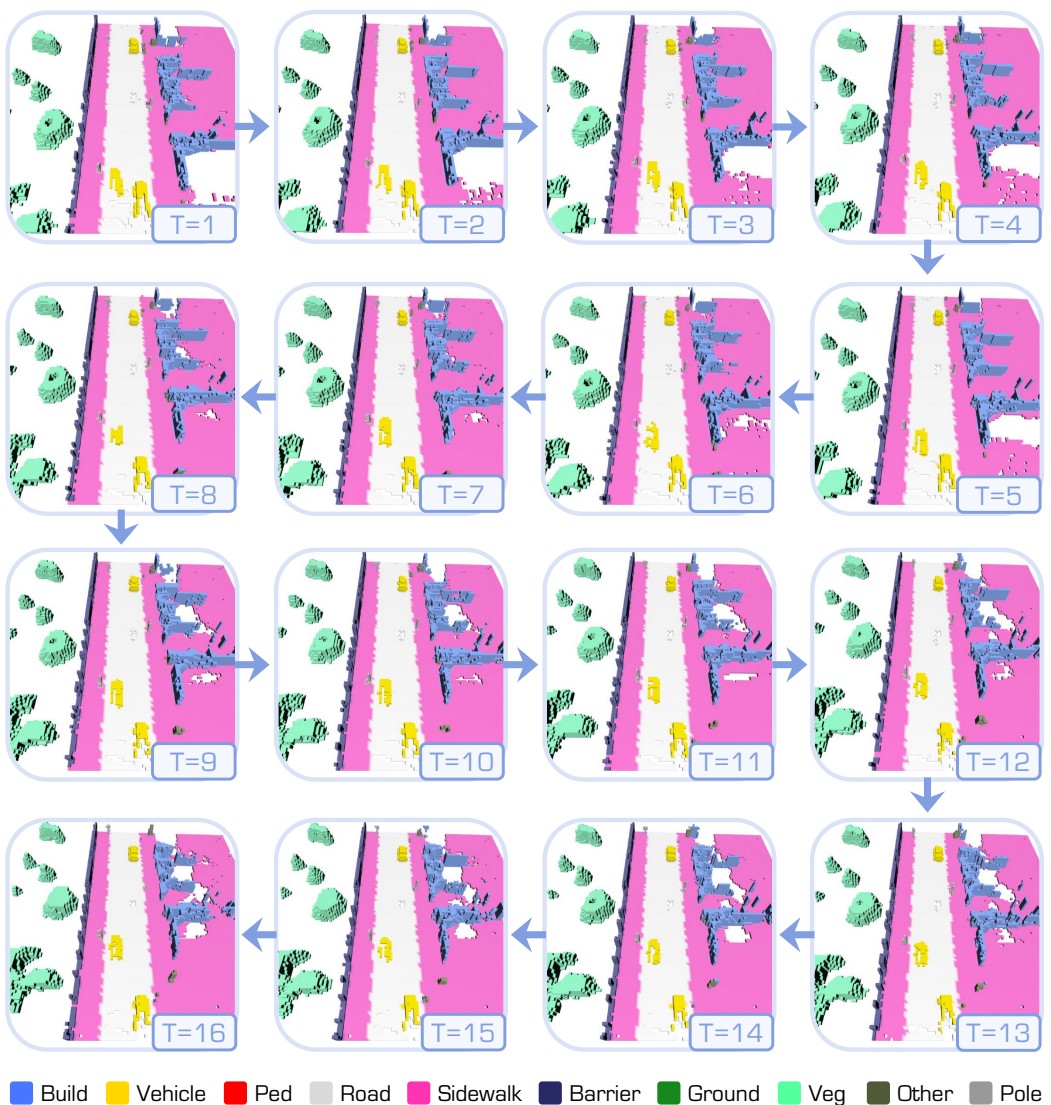

Figure 10: **Unconditional Dynamic Scene Generation Results.** We provide qualitative examples of a total of 16 consecutive frames generated by DynamicCity on the *CarlaSC* (Wilson et al., 2022) dataset. Best viewed in colors and zoomed-in for additional details.

## C.2 HEXPLANE-GUIDED GENERATION

We show results for our HexPlane conditional generation in Fig. 11. Although the sequences are generated in groups of 16 due to the settings of our VAE, we successfully generate a long sequence by conditioning on the previous one. The result contains 64 frames, comprising four sequences, and depicts a T-intersection with many cars parked along the roadside. This result demonstrates strong temporal consistency across sequences, proving that our framework can effectively predict the next sequence based on the current one.

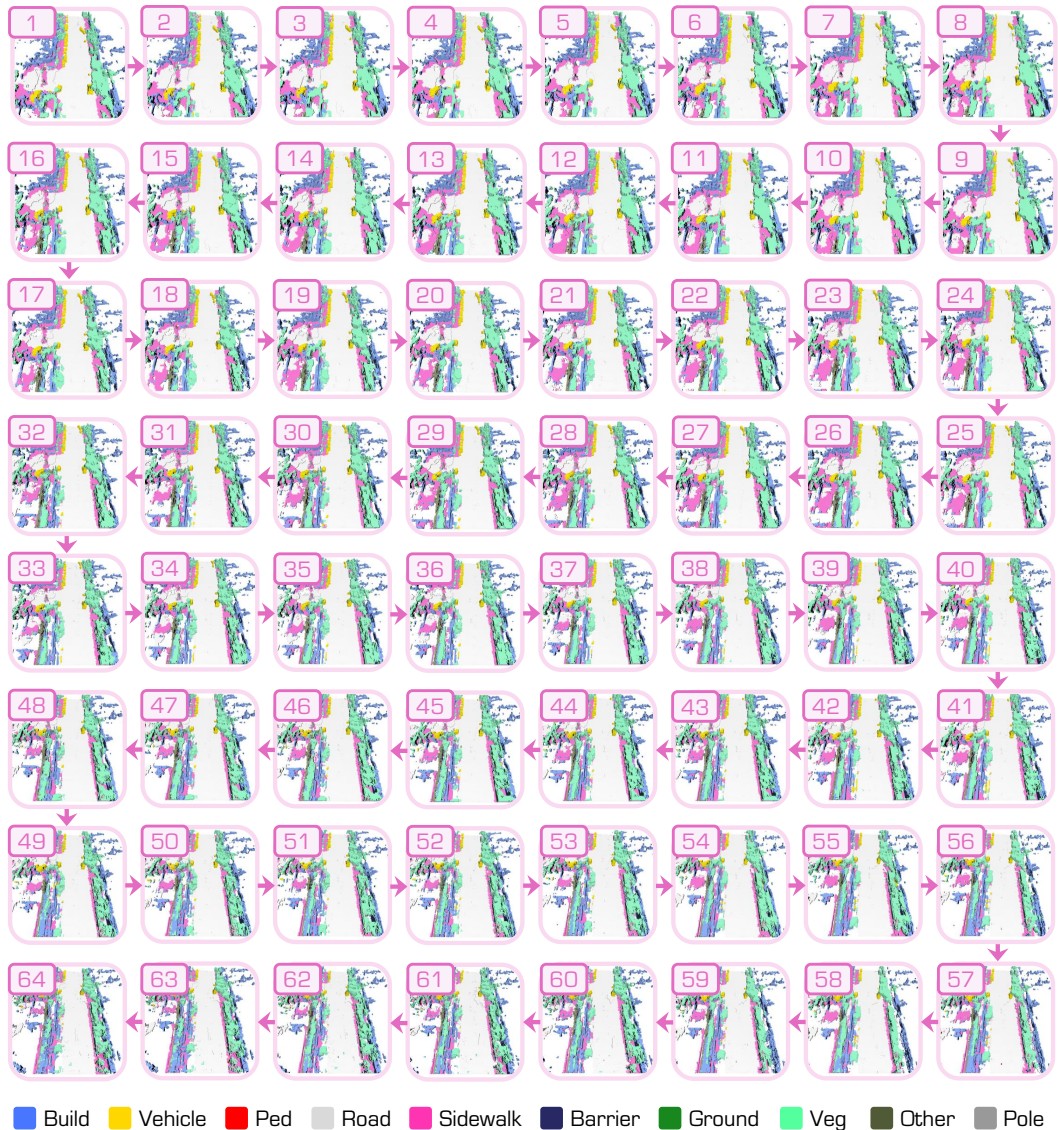

Figure 11: **HexPlane-Guided Generation Results.** We provide qualitative examples of a total of 64 consectutive frames generated by DynamicCity on the *Occ3D-Waymo* (Tian et al., 2023) dataset. Best viewed in colors and zoomed-in for additional details.

## C.3 LAYOUT-GUIDED GENERATION

The layout conditional generation result is presented in Fig. 12. First, we observe that the layout closely matches the semantic positions in the generated result. Additionally, as the layout changes, the positions of the vehicles in the scene also change accordingly, demonstrating that our model effectively captures the condition and influences both the overall scene layout and vehicle placement.

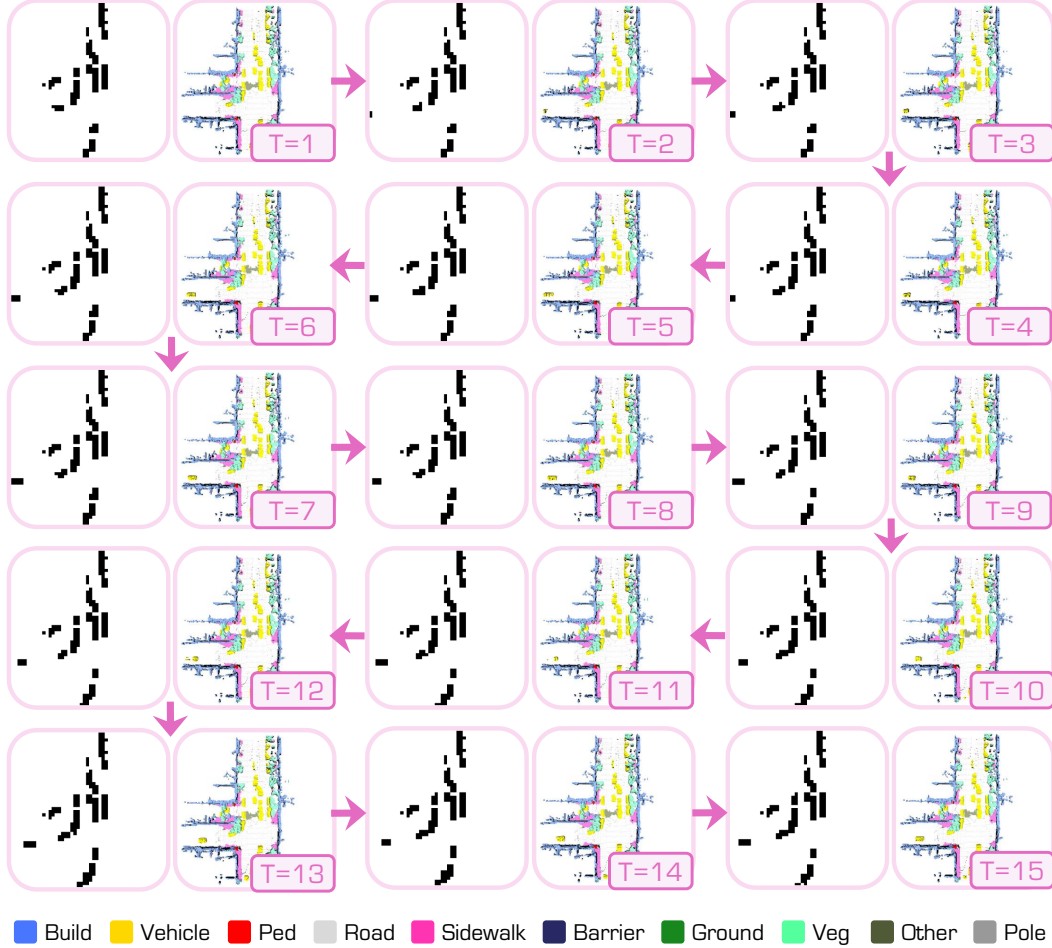

Figure 12: **Layout-Guided Generation Results.** We provide qualitative examples of a total of 16 consecutive frames generated by DynamicCity on the *Occ3D-Waymo* (Tian et al., 2023) dataset. Best viewed in colors and zoomed-in for additional details.

## C.4 COMMAND- & TRAJECTORY-GUIDED GENERATION

We present command conditional generation in Fig. 13 and trajectory conditional generation in Fig. 14. These results show that when we input a command, such as "right turn," or a sequence of XY-plane coordinates, our model can effectively control the motion of the ego vehicle and the relative motion of the entire scene based on these movement trends.

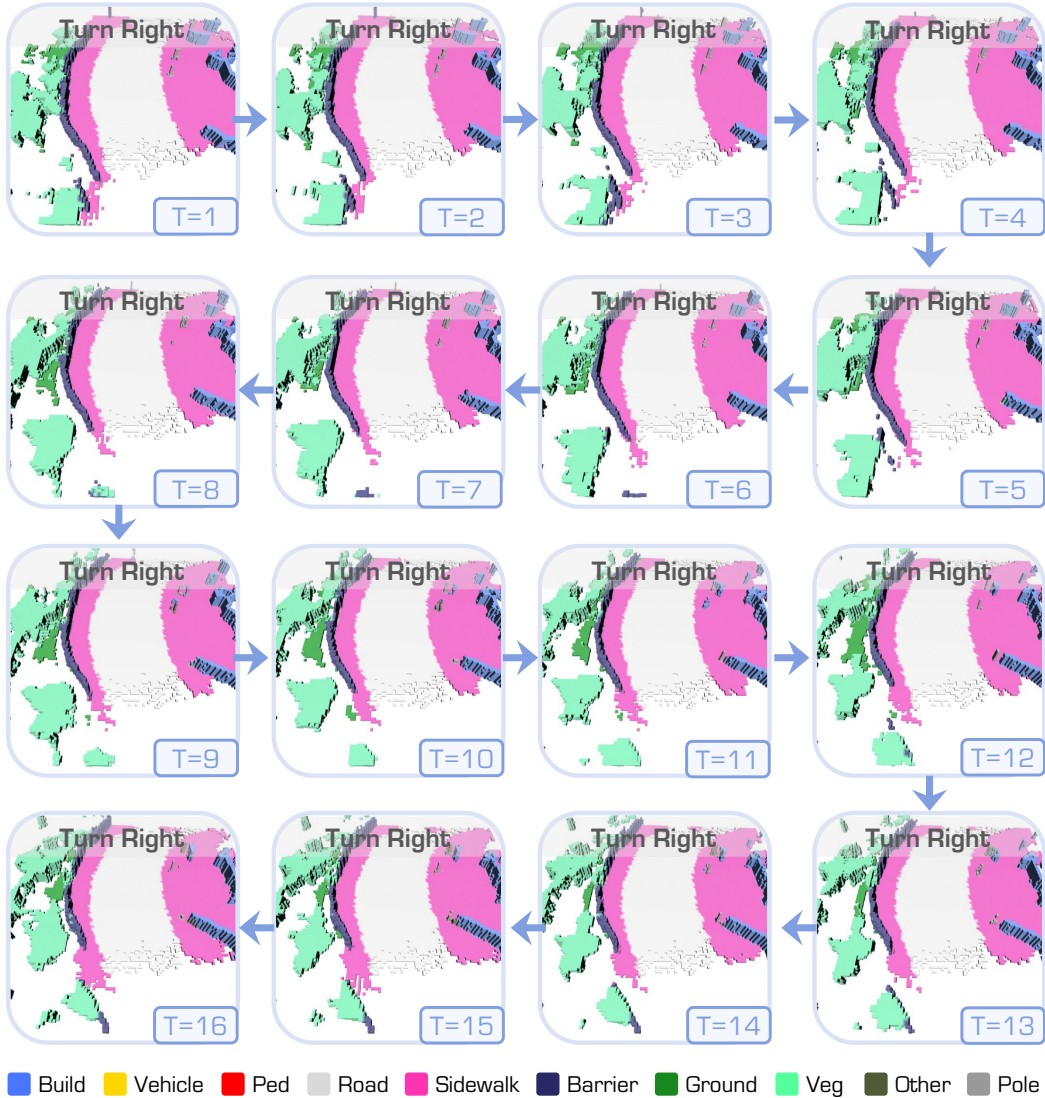

Figure 13: **Command-Guided Scene Generation Results.** We provide qualitative examples of a total of 16 consecutive frames generated under the command RIGHT by DynamicCity on the *CarlaSC* (Wilson et al., 2022) dataset. Best viewed in colors and zoomed-in for additional details.

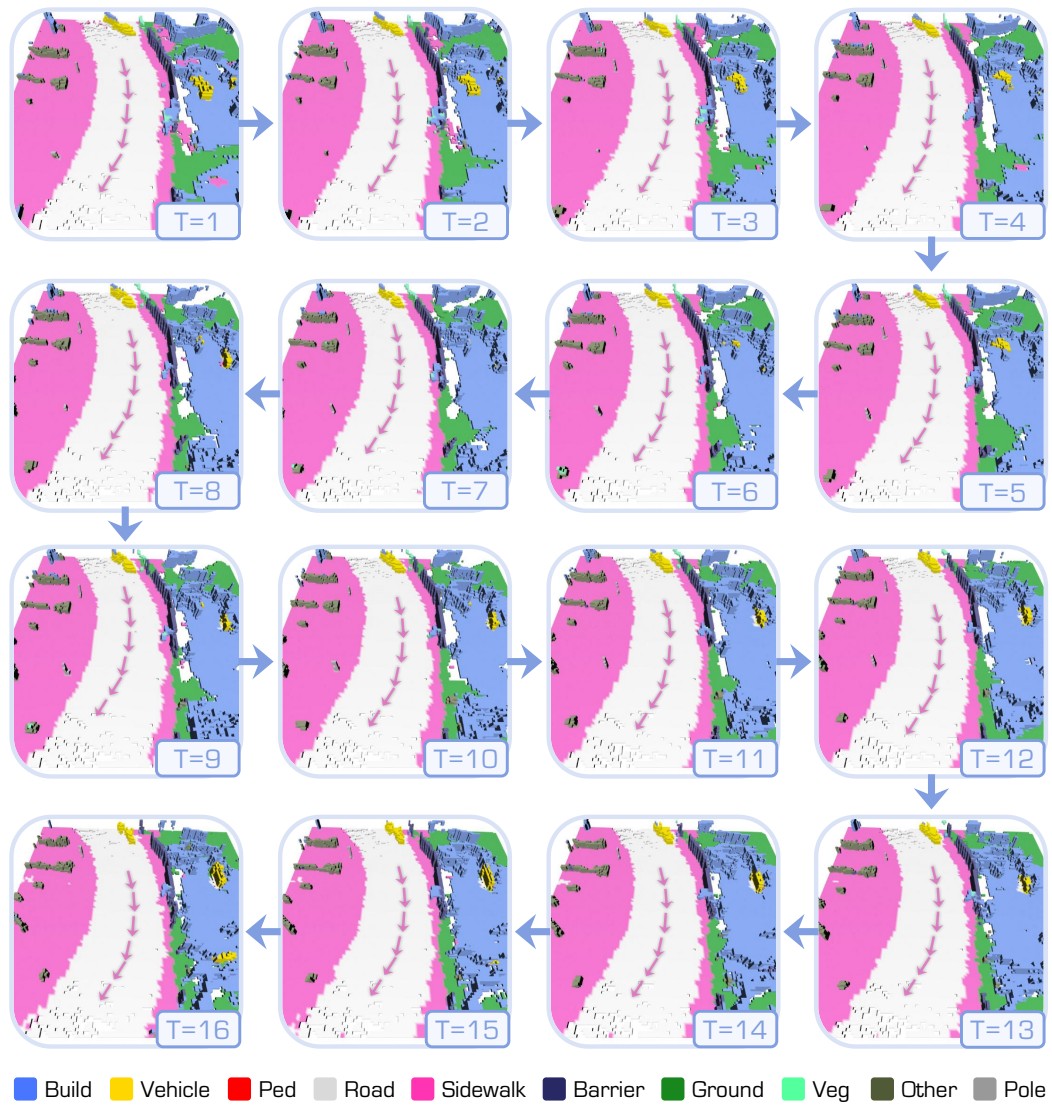

Figure 14: **Trajectory-Guided Scene Generation Results.** We provide qualitative examples of a total of 16 consecutive frames generated by DynamicCity on the *CarlaSC* (Wilson et al., 2022) dataset. Best viewed in colors and zoomed-in for additional details.

## C.5 DYNAMIC INPAINTING

We present the full inpainting results in Fig. 15. The results show that our model successfully regenerates the inpainted regions while ensuring that the areas outside the inpainted regions remain consistent with the original scene. Furthermore, the inpainted areas seamlessly blend into the original scene, exhibiting realistic placement and dynamics.

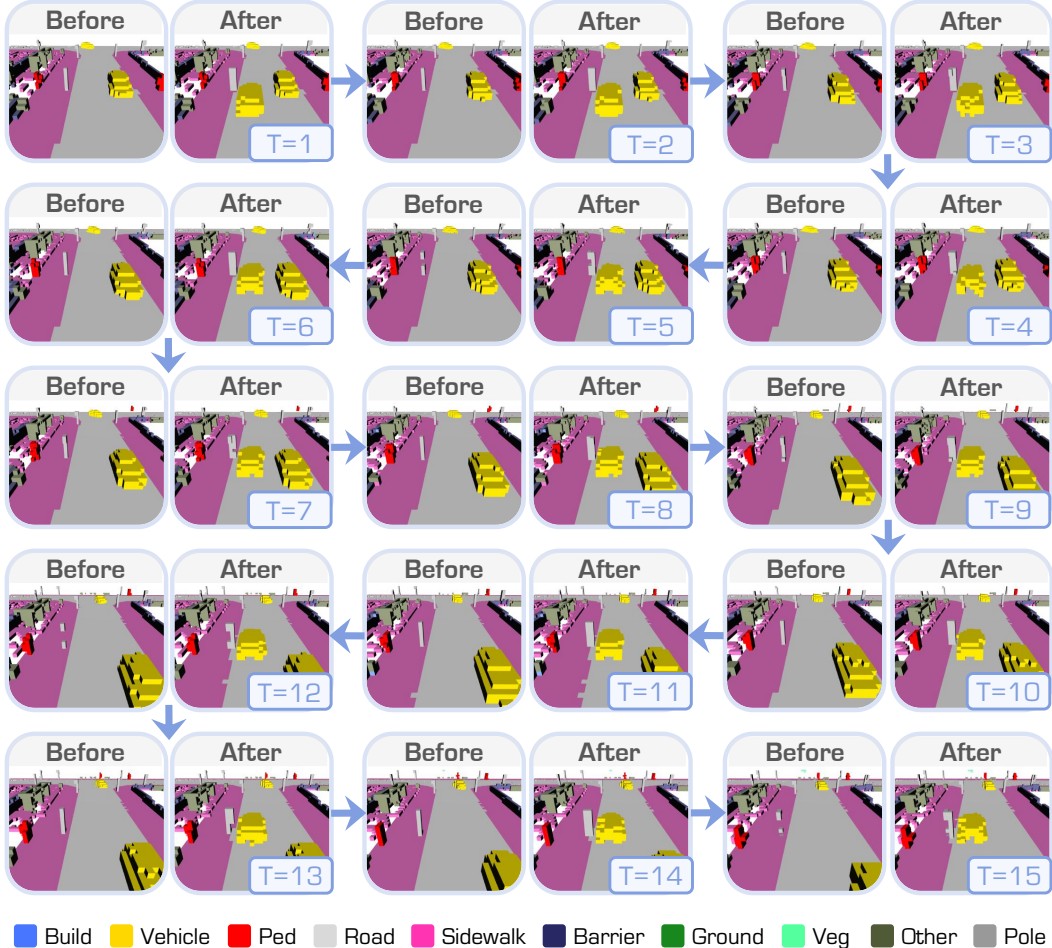

Figure 15: **Dynamic Inpainting Results.** We provide qualitative examples of a total of 16 consecutive frames generated by DynamicCity on the *CarlaSC* (Wilson et al., 2022) dataset. Best viewed in colors and zoomed-in for additional details.

## C.6 COMPARISONS WITH OCCSORA

We compare our qualitative results with OccSora (Wang et al., 2024) in Fig. 16, using a similar scene. It is evident that our result presents a realistic dynamic scene, with straight roads and complete objects and environments. In contrast, OccSora's result displays unreasonable semantics, such as a pedestrian in the middle of the road, broken vehicles, and a lack of dynamic elements. This comparison highlights the effectiveness of our method.

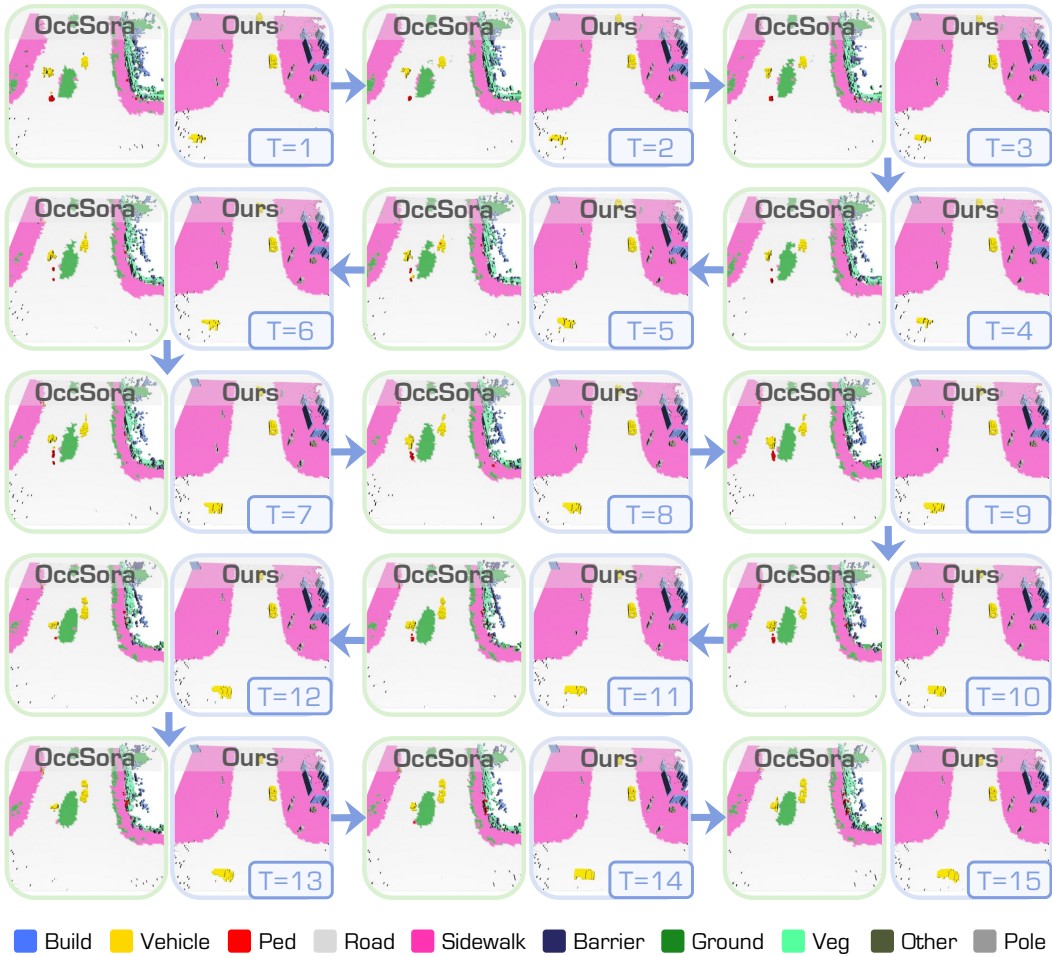

Figure 16: **Comparisons of Dynamic Scene Generation.** We provide qualitative examples of a total of 16 consecutive frames generated by OccSora (Wang et al., 2024) and our proposed DynamicCity framework on the *CarlaSC* (Wilson et al., 2022) dataset. Best viewed in colors and zoomed-in for additional details.

## C.7 DYNAMIC OUTPAINTING

We present the full outpainting results in Fig. 17. The results demonstrate that our model can extend a scene into a larger dynamic scene.

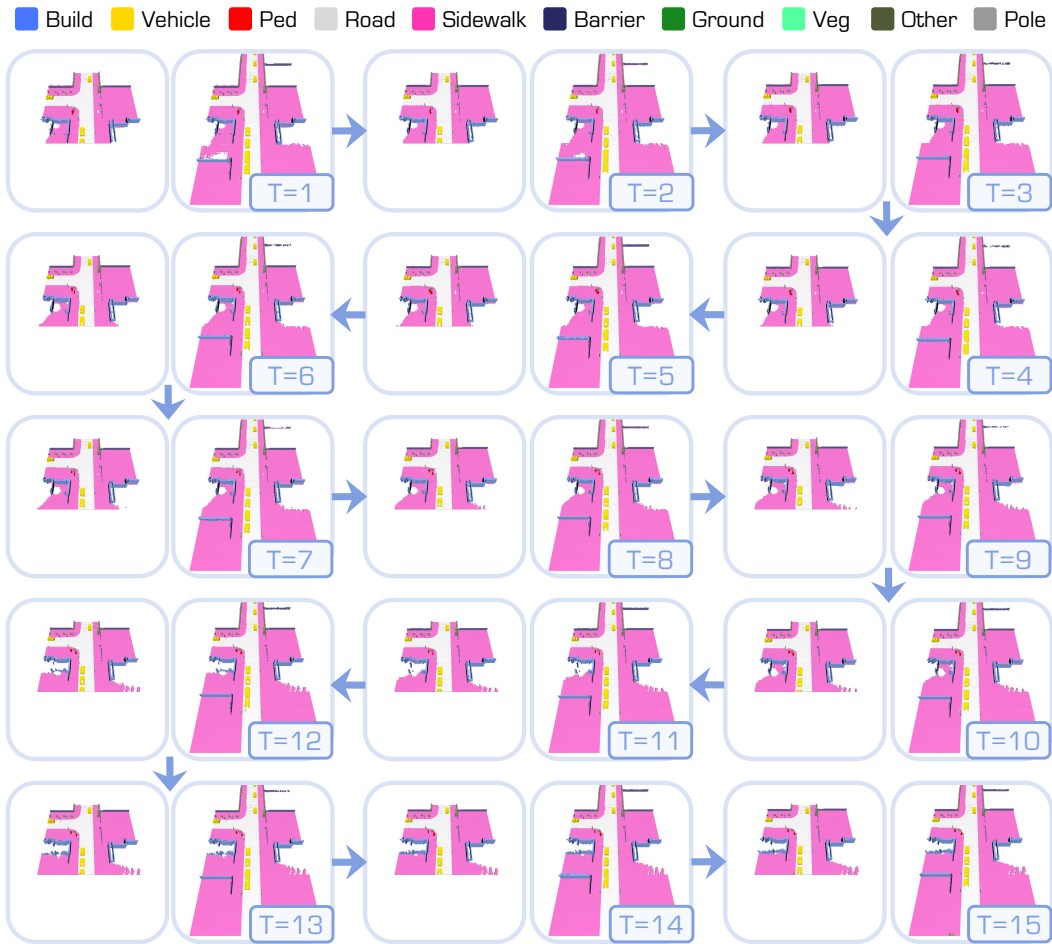

Figure 17: **Dynamic Outpainting Results.** We provide qualitative examples of a total of 16 consecutive frames generated by DynamicCity on the *CarlaSC* (Wilson et al., 2022) dataset. Best viewed in colors and zoomed-in for additional details.

## C.8 SINGLE FRAME OCCUPANCY CONDITIONAL GENERATION

We present the results of generating frames based on a single-frame occupancy condition in Fig. 18. The results demonstrate good temporal consistency with the condition frame, highlighting our model's ability to condition on easily accessible data.

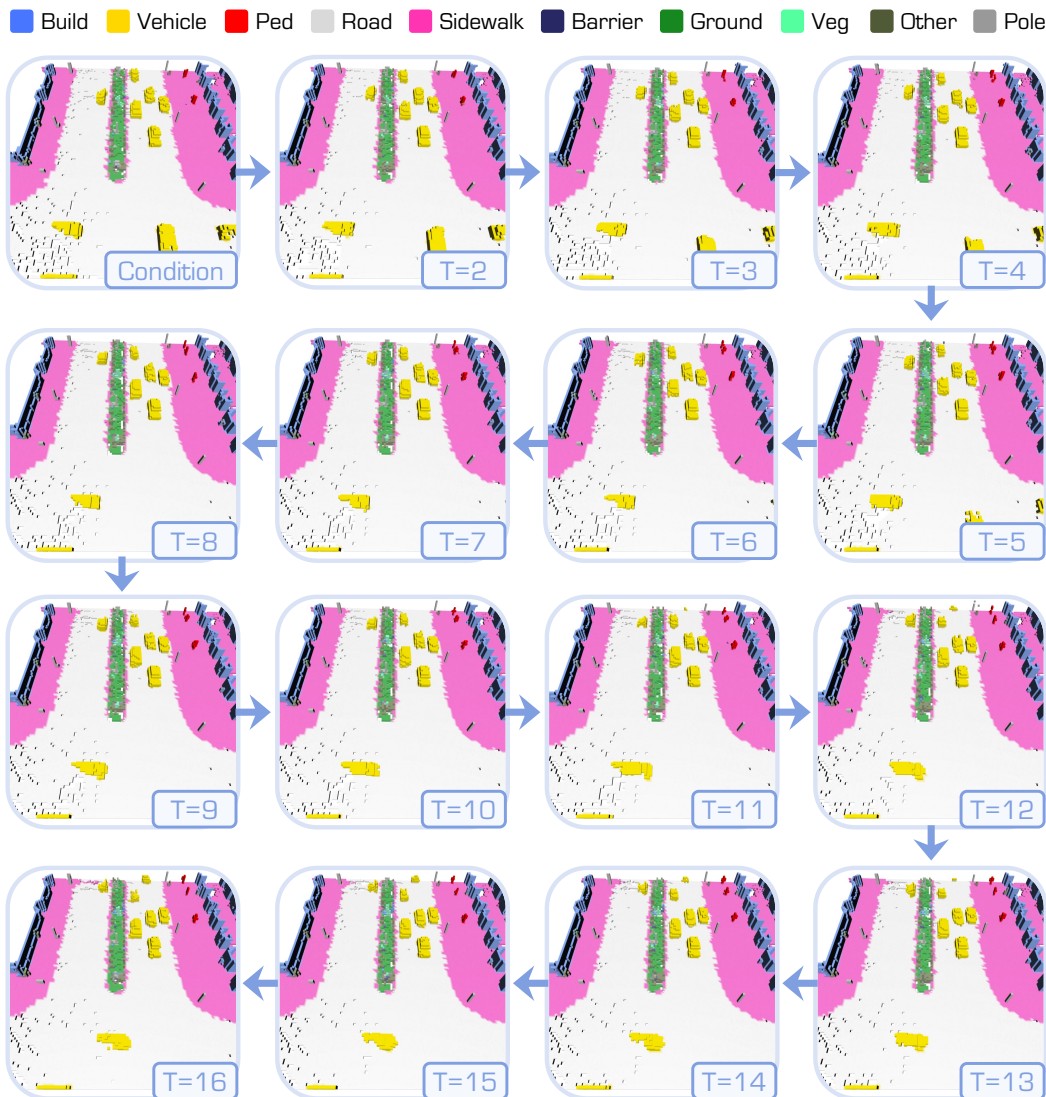

Figure 18: **Single Frame Occupancy Conditional Generation Results.** We provide qualitative examples of a total of 16 consecutive frames generated by DynamicCity on the *CarlaSC* (Wilson et al., 2022) dataset. Best viewed in colors and zoomed-in for additional details.

# D  POTENTIAL SOCIETAL IMPACT & LIMITATIONS

In this section, we elaborate on the potential positive and negative societal impact of this work, as well as the broader impact and some potential limitations.

## D.1  SOCIETAL IMPACT

Our approach's ability to generate high-quality 4D occupancy holds the potential to significantly impact various domains, particularly autonomous driving, robotics, urban planning, and smart city development. By creating realistic, large-scale dynamic scenes, our model can aid in developing more robust and safe autonomous systems. These systems can be better trained and evaluated against diverse scenarios, including rare but critical edge cases like unexpected pedestrian movements or complex traffic patterns, which are difficult to capture in real-world datasets. This contribution can lead to safer autonomous vehicles, reducing traffic accidents, and improving traffic efficiency, ultimately benefiting society by enhancing transportation systems.

In addition to autonomous driving, DynamicCity can be valuable for developing virtual reality (VR) environments and augmented reality (AR) applications, enabling more realistic 3D simulations that could be used in various industries, including entertainment, training, and education. These advancements could help improve skill development in driving schools, emergency response training, and urban planning scenarios, fostering a safer and more informed society.

Despite these positive outcomes, the technology could be misused. The ability to generate realistic dynamic scenes might be exploited to create misleading or fake data, potentially undermining trust in autonomous systems or spreading misinformation about the capabilities of such technologies. However, we do not foresee any direct harmful impact from the intended use of this work, and ethical guidelines and responsible practices can mitigate potential risks.

## D.2  BROADER IMPACT

Our approach's contribution to 4D scene generation stands to advance the fields of autonomous driving, robotics, and even urban planning. By providing a scalable solution for generating diverse and dynamic scenes, it enables researchers and engineers to develop more sophisticated models capable of handling real-world complexity. This has the potential to accelerate progress in autonomous systems, making them safer, more reliable, and adaptable to a wide range of environments. For example, researchers can use DynamicCity to generate synthetic training data, supplementing real-world data, which is often expensive and time-consuming to collect, especially in dynamic and high-risk scenarios.

The broader impact also extends to lowering entry barriers for smaller research institutions and startups that may not have access to vast amounts of real-world occupancy data. By offering a means to generate realistic and dynamic scenes, DynamicCity democratizes access to high-quality data for training and validating machine learning models, thereby fostering innovation across the autonomous driving and robotics communities.

However, it is crucial to emphasize that synthetic data should be used responsibly. As our model generates highly realistic scenes, there is a risk that reliance on synthetic data could lead to models that fail to generalize effectively in real-world settings, especially if the generated scenes do not capture the full diversity or rare conditions found in real environments. Hence, it's important to complement synthetic data with real-world data and ensure transparency when using synthetic data in model training and evaluation.

## D.3  KNOWN LIMITATIONS

Despite the strengths of DynamicCity, several limitations should be acknowledged. First, our model's ability to generate extremely long sequences is still constrained by computational resources, leading to potential challenges in accurately modeling scenarios that span extensive periods. While we employ techniques to extend temporal modeling, there may be degradation in scene quality or consistency when attempting to generate sequences beyond a certain length, particularly in complex traffic scenarios.

Second, the generalization capability of DynamicCity depends on the diversity and representativeness of the training datasets. If the training data does not cover certain environmental conditions, object categories, or dynamic behaviors, the generated scenes might lack these aspects, resulting in incomplete or less realistic dynamic occupancy data. This could limit the model's effectiveness in handling unseen or rare scenarios, which are critical for validating the robustness of autonomous systems.

Third, while our model demonstrates strong performance in generating dynamic scenes, it may face challenges in highly congested or intricate traffic environments, where multiple objects interact closely with rapid, unpredictable movements. In such cases, DynamicCity might struggle to capture the fine-grained details and interactions accurately, leading to less realistic scene generation.

Lastly, the reliance on pre-defined semantic categories means that any variations or new object types not included in the training set might be inadequately represented in the generated scenes. Addressing these limitations would require integrating more diverse training data, improving the model's adaptability, and refining techniques for longer sequence generation.

# E    PUBLIC RESOURCES USED

In this section, we acknowledge the public resources used, during the course of this work.

## E.1    PUBLIC DATASETS USED

- nuScenes[1] ................................................ CC BY-NC-SA 4.0
- nuScenes-devkit[2] ............................................ Apache License 2.0
- Waymo Open Dataset[3] .................................... Waymo Dataset License
- CarlaSC[4] ........................................................ MIT License
- Occ3D[5] ......................................................... MIT License

## E.2    PUBLIC IMPLEMENTATIONS USED

- SemCity[6] ........................................................ Unknown
- OccSora[7] .................................................. Apache License 2.0
- MinkowskiEngine[8] ............................................... MIT License
- TorchSparse[9] ................................................... MIT License
- SPVNAS[10] ...................................................... MIT License
- spconv[11] .................................................. Apache License 2.0

---

[1] https://www.nuscenes.org/nuscenes
[2] https://github.com/nutonomy/nuscenes-devkit
[3] https://waymo.com/open
[4] https://umich-curly.github.io/CarlaSC.github.io.
[5] https://tsinghua-mars-lab.github.io/Occ3D.
[6] https://github.com/zoomin-lee/SemCity.
[7] https://github.com/wzzheng/OccSora.
[8] https://github.com/NVIDIA/MinkowskiEngine.
[9] https://github.com/mit-han-lab/torchsparse.
[10] https://github.com/mit-han-lab/spvnas.
[11] https://github.com/traveller59/spconv.

