# OpenReview forum: "DynamicCity: Large-Scale 4D Occupancy Generation from Dynamic Scenes"
_ICLR.cc/2025/Conference — ICLR 2025 Spotlight_

### Official Review · Reviewer_URnX · 2024-10-28

**Soundness:** 3
**Presentation:** 3
**Contribution:** 3
**Rating:** 6
**Confidence:** 2

**Summary:**

This paper proposes DynamicCity for large-scale driving environments for 4D generation. The paper proposes a Projection Module to efficiently 4D features into six 2D feature maps and an Expansion & Squeeze Strategy to reconstruct 3D feature volumes in parallel. A Padded Rollout Operation is also proposed to reorganize all six feature planes of the HexPlane as a squared 2D feature map.

**Strengths:**

1. The writing is easy to follow.

2. The ablation is comprehensive and validates the efficiency of Padded Rollout, Projection Module, and ESS in Table 3 and Table 5.

3. The results are good compared to the baseline method over different datasets.

4. The trajectory-guided generation, dynamic scene inpainting, and layout-conditioned generation show various downstream applications.

**Weaknesses:**

1. The Dynamic Object Inpainting Results seem to make the ground have holes(Figure 1). Could you please explain this phenomenon and discuss any potential limitations or artifacts of this inpainting approach?

2. The results seem to reflect semantic occupancy, with training data voxelized into occupancy format using LiDAR point clouds. Could you clarify why "LiDAR generation" is used in the title, abstract, and introduction? It would be helpful to explain the distinction between "LiDAR generation" and "semantic occupancy generation" as used throughout the paper, as well as the relationship between LiDAR data and the voxelized occupancy representation. This clarification would help ensure that the terminology accurately represents the method and results.

3. It’s recommended to add a comparison of model size across different methods, as the proposed approach appears to require substantial training resources.

4. While a training time comparison is provided in the ablation study, it’s still suggested to include a comparison of training time and GPU requirements specifically against the baseline method, like Occ-sora. These comparisons could help readers better understand how much computational power the proposed method needs compared to other established methods.

**Questions:**

Same as Weakness.

---

> ### Author Response · Authors · 2024-11-23
> **Response to Reviewer URnX**
>
> We sincerely thank Reviewer `URnX` for the valuable comments provided during this review. The point-to-point responses are as follows.
>
> ---
>
> > **W1:** *"The Dynamic Object Inpainting Results seem to make the ground have holes (Fig. 1). Could you please explain this phenomenon and discuss any potential limitations or artifacts of this inpainting approach?"*
>
> **A:** Thanks for your comment. We would like to clarify that the "holes" on the ground you mentioned are part of the input ground truth occupancy to the inpainting. While our inpainting results may occasionally exhibit small holes on the ground, we believe this reflects adherence to the distribution of the training data.
>
> ---
>
> > **W2:** *"It would be helpful to explain the distinction between 'LiDAR generation' and 'semantic occupancy generation'."*
>
> **A:** Thank you for your valuable feedback. To avoid ambiguity, we have updated the "LiDAR" in the title of our paper to "Occupancy". Occupancy usually represents concatenated dense LiDAR scenes, and we have added more necessary explanations in our revised paper.
>
> ---
>
> > **W3:** *"It’s recommended to compare model sizes across different methods."*
>
> **A:** Thanks for your suggestion. Below, we have included a comparison of model size between our method and OccSora. Our models are significantly smaller than that of OccSora. For other comparisons refer to Table 2 to Table 5 below.
>
> **Table 1:** Model Size Comparison
> | Dataset | Method | Model Size (M) |
> |:-:|:-:|-:|
> | CarlaSC     | OccSora  |169 |
> | CarlaSC     | Ours     | 44 |
> | Occ3D-Waymo | OccSora  |174 |
> | Occ3D-Waymo | Ours     | 45 |
>
> ---
>
> > **W4:** *"It’s suggested to include a comparison of training time and GPU requirements against the baseline method."*
>
> **A:** Thanks for your comment. Below, we have included a comparison of GPU memory usage, training and inference time between our method and OccSora. While our method can be slightly slower and consumes more memory compared to OccSora, we achieve much better performance.
>
> **Table 2:** Autoencoder on CarlaSC
> |Method|Training Time|Inference Time|Training Memory|Inference Memory|
> |:-:|:-:|:-:|:-:|:-:|
> |OccSora|0.36|0.21|4.86|3.25|
> |Ours|0.21|0.41|5.92|1.43|
>
> **Table 3:** Autoencoder on Occ3D-Waymo
> |Method|Training Time|Inference Time|Training Memory|Inference Memory|
> |:-:|:-:|:-:|:-:|:-:|
> |OccSora|0.63|0.21|10.05|3.93|
> |Ours|0.94|0.54|24.90|4.62|
>
> **Table 4:** DiT on CarlaSC
> |Method|Training Time|Inference Time|Training Memory|Inference Memory|
> |:-:|:-:|:-:|:-:|:-:|
> |OccSora|0.19|6.10|1.50|1.15|
> |Ours|0.19|3.91|10.22|1.28|
>
> **Table 5:** DiT on Occ3D-Waymo
> |Method|Training Time|Inference Time|Training Memory|Inference Memory|
> |:-:|:-:|:-:|:-:|:-:|
> |OccSora|0.35|6.09|15.16|1.15|
> |Ours|0.45|4.41|22.33|1.29|
>
> ---
>
> Last but not least, we sincerely thank Reviewer `URnX` again for the valuable comments provided during this review.

---

> > ### Author Response · Authors · 2024-11-25
> > **Looking forward to hearing from you**
> >
> > **Dear Reviewer `URnX`,**
> >
> > We sincerely appreciate your time and effort in reviewing our manuscript and providing valuable feedback.
> >
> > ---
> >
> > In response to your insightful comments, we have made the following revisions to our manuscript and project page:
> >
> > - We have changed the "LiDAR" in our title to "Occupancy" and added additional explanations to avoid ambiguity.
> > - We have updated the 2D generation evaluation results in Table 2 and included an example of the 2D semantic map in Appendix Sec. A.2.
> > - We have added the occupancy forecasting result in Appendix Sec. B.2.
> > - We have added user study results, including the evaluation of temporal consistency, in Appendix Sec. B.3.
> > - We have added a comparison of model speed, memory consumption, and size with OccSora in Appendix Sec. B.4.
> > - We have included a comparison with SemCity in Appendix Sec. B.5.
> > - We have added outpainting results in Appendix Sec. C.7.
> > - We have added single-frame occupancy conditional generation results in Appendix Sec. C.8.
> > - We have fixed typos and formatting errors.
> >
> > We hope these revisions adequately address your concerns.
> >
> > ---
> >
> > We look forward to actively participating in the **Author-Reviewer Discussion** session and welcome any additional feedback you might have on the manuscript or the changes we have made.
> >
> > ---
> >
> > Once again, we sincerely thank you for your contributions to improving the quality of this work.
> >
> > Best regards,
> >
> > The Authors of Submission 552

---

> > > ### Author Response · Authors · 2024-11-28
> > > **Looking forward to hearing from you**
> > >
> > > **Dear Reviewer `URnX`,**
> > >
> > > We sincerely appreciate your time and effort in reviewing our manuscript and providing valuable feedback.
> > >
> > > ---
> > >
> > > We have made corresponding revisions based on your insightful comments. Please let us know whether these responses address your concerns or if there are additional aspects that we can look at to improve this work further.
> > >
> > > ---
> > >
> > > We really cherish the **Author-Reviewer Discussion** session and welcome any additional feedback you might have on the manuscript or the changes we have made.
> > >
> > >
> > > Best regards,
> > >
> > > The Authors of Submission 552

---

> ### Comment · Reviewer_URnX · 2024-11-29
>
> Dear authors,
>
> I'm glad to observe the improvement of this paper. Actually, although I'm familiar with the street scale problem, I'm not so professional in the dynamic problem.
>
> I take time to review your paper and give responsible, but not so much breaking feedback that I'm sure won't harm this paper.
>
> Since I'm not so professional in dynamic problems, I may not be able or qualified to judge whether to give a stronger acceptance recommendation based on my knowledge.
>
> Considering this, I lower my confidence.
>
> **Update:**
>
> Sorry that I forget to choose the authors to be visible in my previous reply:
>
> > I acknowledge the authors' reply. My concerns have been addressed. I keep my positive rating.
>
> But I'm still deciding to lower my confidence after consideration for responsible feedback.

---

> > ### Author Response · Authors · 2024-11-29
> >
> > **Dear Reviewer `URnX`,**
> >
> > Thank you for your thoughtful review and constructive feedback. We’re glad our revisions addressed your concerns and appreciate your positive rating. Your insights have helped us refine our approach and presentation.
> >
> > ---
> >
> > Regarding your note on confidence, we fully respect and understand your position. We truly appreciate your effort in thoroughly considering our work, even for aspects outside your immediate expertise. If there are any additional details or clarifications that could assist you further in evaluating the dynamic problem addressed in our work, please don’t hesitate to let us know.
> >
> > ---
> >
> > Thank you again for your thoughtful review and your constructive engagement with our research.
> >
> > Best regards,
> >
> > The Authors of Submission 552

---

### Official Review · Reviewer_a6Q2 · 2024-11-03

**Soundness:** 3
**Presentation:** 4
**Contribution:** 3
**Rating:** 8
**Confidence:** 4

**Summary:**

This paper presents DynamicCity, a novel 4D scene generation method for driving environments. DynamicCity utilizes a VAE model to compress 4D scenes into a HexPlane, integrating strategies such as a Projection Module and an Expansion & Squeeze Strategy to enhance performance and efficiency. A DiT-based diffusion model is then employed for generating the HexPlane. The method supports various 4D scene generation applications, including trajectory, command, and layout conditions. Basically this is a strong submission with impressive results, while I have several further concerns.

**Strengths:**

1. The paper is well-structured and easy to follow.
2. The generated results are impressive, with an accompanying demonstration video that effectively showcases the method’s capabilities.
3. The proposed method has a clear motivation and presents a well-reasoned pipeline.

**Weaknesses:**

1. I’m a little confused by the title and the task definition. While the authors state that the proposed method is designed for “LiDAR” generation, the results seem more akin to “occupancy” generation. These concepts are distinct, despite both representing the scene’s geometry. For true LiDAR generation [1][2], the outputs should be LiDAR point clouds that reflect the sampling properties of LiDAR sensors (e.g., ray drop, ray-based sampling, etc).
2. The authors note that the method can support long sequential modeling of up to 128 frames, but the factors limiting sequence length (e.g., GPU memory) are not discussed. Additionally, the inference running time is not mentioned.
3. Can the proposed method support outpainting similar to SemCity? While inpainting results are shown to demonstrate the spatial significance of the HexPlane, outpainting results could further demonstrate the method’s ability to expand scenes. Furthermore, the inpainting examples in the paper are confined to small regions.
4. Regarding the experimental results, I suggest to provide quantitative comparison with SemCity in static scenes. The caption of Table 2 points out the authors compare the method with SemCIty, but I do not see results in the table.

[1] Ran, Haoxi, Vitor Guizilini, and Yue Wang. "Towards Realistic Scene Generation with LiDAR Diffusion Models." *Proceedings of the IEEE/CVF Conference on Computer Vision and Pattern Recognition*. 2024.

[2] Zyrianov, Vlas, et al. "LidarDM: Generative LiDAR Simulation in a Generated World." *arXiv preprint arXiv:2404.02903* (2024).

**Questions:**

Please refer to the Weaknesses.

---

> ### Author Response · Authors · 2024-11-23
> **Response to Reviewer a6Q2**
>
> We sincerely thank Reviewer `a6Q2` for the valuable comments provided during this review. The point-to-point responses are as follows.
>
> ---
>
> > **W1:** *"... confused by the title ... “occupancy” generation ..."*
>
> **A:** Thank you for your valuable feedback. To avoid ambiguity, we have changed the term "LiDAR" in our title to "Occupancy". We add the necessary explanations in the revised paper.
>
> ---
>
> > **W2:** *"... method can support long sequential modeling of up to 128 frames, but the factors limiting sequence length (e.g., GPU memory) are not discussed. Additionally, the inference running time is not mentioned."*
>
> **A:** Thanks for your comment. Actually, our long sequential modeling is not limited by GPU memory. It is achieved through auto-regressively generating HexPlanes conditioned on the previous HexPlane. Since we perform long-sequence generation auto-regressively, HexPlanes are generated one at a time, and GPU memory consumption remains similar to that of unconditional generation. We include statistics such as GPU memory usage and inference running time in the table below. While some of our models can be slightly slower and consumes more memory compared with OccSora, we achieves much better performance.
>
> **Table 1:** Autoencoder on CarlaSC
> |Method|Training Time|Inference Time|Training Memory|Inference Memory|
> |:-:|:-:|:-:|:-:|:-:|
> |OccSora|0.36|0.21|4.86|3.25|
> |Ours|0.21|0.41|5.92|1.43|
>
> **Table 2:** Autoencoder on Occ3D-Waymo
> |Method|Training Time|Inference Time|Training Memory|Inference Memory|
> |:-:|:-:|:-:|:-:|:-:|
> |OccSora|0.63|0.21|10.05|3.93|
> |Ours|0.94|0.54|24.90|4.62|
>
> **Table 3:** DiT on CarlaSC
> |Method|Training Time|Inference Time|Training Memory|Inference Memory|
> |:-:|:-:|:-:|:-:|:-:|
> |OccSora|0.19|6.10|1.50|1.15|
> |Ours|0.19|3.91|10.22|1.28|
>
> **Table 4:** DiT on Occ3D-Waymo
> |Method|Training Time|Inference Time|Training Memory |Inference Memory|
> |:-:|:-:|:-:|:-:|:-:|
> |OccSora|0.35|6.09|15.16|1.15|
> |Ours|0.45|4.41|22.33|1.29|
>
> The tables have been included in Appendix Section B.4. Theoretically, our model can generate sequence with arbitrary length, but less stable generation may occur when generating extremely long sequences. Hence, we report "up to 128 frames" in our paper.
>
> ---
>
> > **W3:** *"Can the proposed method support outpainting similar to SemCity? ... Furthermore, the inpainting examples in the paper are confined to small regions."*
>
> **A:** Thanks for your question.
>
> - **Outpainting.** Our model can support outpainting, and outpainting results have been included on our website and Appendix C.7 to demonstrate its ability to expand scenes.
> - **Inpainting.** We would like to clarify that the region of inpainting depends on the given mask is not restricted to small regions. To illustrate this, we provide more inpainting results with much larger inpainted regions, observed from a different perspective, on our [website](https://dynamic-city.github.io/#5-dynamic-scene-inpainting).
>
> ---
>
> > **W4:** *"... to provide quantitative comparison with SemCity in static scenes. The caption of Table 2 points out the authors compare the method with SemCIty, but I do not see results in the table."*
>
> **A:** Thanks for your comment. "SemCity" in Table 2 is a typo and has been removed in the revised PDF. Table 2 in our paper is intended to compare 4D generation results, therefore we did not include SemCity, which is a 3D generation method. We take your suggestion and conduct a quantitative comparison of our method against SemCity on the CarlaSC dataset. The results are as follows:
>
> |Method|IS-2D|FID-2D|KID-2D|P-2D|R-2D|IS-3D|FID-3D| KID-3D|P-3D|R-3D|
> |:-:|:-:|:-:|:-:|:-:|:-:|:-:|:-:|:-:|:-:|:-:|
> |SemCity| 1.039|35.40|0.010|0.213|0.058|2.288|1113| 53.948|0.253|0.787|
> |Ours|1.040|12.94|0.002|0.307|0.018|2.331|427.5| 27.869|0.460|0.170|
>
> Details can be found in the Appendix. B.5.
>
> ---
>
> Last but not least, we sincerely thank Reviewer `a6Q2` again for the valuable comments provided during this review.

---

> > ### Author Response · Authors · 2024-11-25
> > **Looking forward to hearing from you**
> >
> > **Dear Reviewer `a6Q2`,**
> >
> > We sincerely appreciate your time and effort in reviewing our manuscript and providing valuable feedback.
> >
> > ---
> >
> > In response to your insightful comments, we have made the following revisions to our manuscript and project page:
> >
> > - We have changed the "LiDAR" in our title to "Occupancy" and added additional explanations to avoid ambiguity.
> > - We have updated the 2D generation evaluation results in Table 2 and included an example of the 2D semantic map in Appendix Sec. A.2.
> > - We have added the occupancy forecasting result in Appendix Sec. B.2.
> > - We have added user study results, including the evaluation of temporal consistency, in Appendix Sec. B.3.
> > - We have added a comparison of model speed, memory consumption, and size with OccSora in Appendix Sec. B.4.
> > - We have included a comparison with SemCity in Appendix Sec. B.5.
> > - We have added outpainting results in Appendix Sec. C.7.
> > - We have added single-frame occupancy conditional generation results in Appendix Sec. C.8.
> > - We have fixed typos and formatting errors.
> >
> > We hope these revisions adequately address your concerns.
> >
> > ---
> >
> > We look forward to actively participating in the **Author-Reviewer Discussion** session and welcome any additional feedback you might have on the manuscript or the changes we have made.
> >
> > ---
> >
> > Once again, we sincerely thank you for your contributions to improving the quality of this work.
> >
> > Best regards,
> >
> > The Authors of Submission 552

---

> > > ### Comment · Reviewer_a6Q2 · 2024-11-25
> > > **Response to Rebuttal**
> > >
> > > I appreciate the authors' detailed rebuttal.
> > > The rebuttal and the additional experimental results have addressed my concerns.
> > > I think this is a strong submission with good results and sufficient contribution.
> > > Thus, I'll keep my positive score as "accept".

---

> > > > ### Author Response · Authors · 2024-11-25
> > > >
> > > > **Dear Reviewer `a6Q2`,**
> > > >
> > > > We are glad to have addressed all your concerns and deeply grateful for your kind recognition of our work. Thank you sincerely for your time and effort in reviewing our submission.

---

### Official Review · Reviewer_nfdL · 2024-11-05

**Soundness:** 3
**Presentation:** 4
**Contribution:** 3
**Rating:** 8
**Confidence:** 3

**Summary:**

The paper proposes a diffusion-based 4D LiDAR scene generation method. This task
is sometimes referred to as "LiDAR world modeling". The approach performs
diffusion in a HexPlane latent space, and the diffusion can be conditioned on
past data in order to perform autoregressive synthesis, or on semantics in order
to perform layout-guided generation. This makes the approach amenable to
closed-loop simulation.

The encoding stage follows a VAE framework, leveraging a LiDAR backbone followed
by a novel projection module which produces spatio-temporal HexPlanes as the
output.

The decoding stage starts with HexPlanes, decodes them into spatio-temporal
feature volumes, and then finally into 4D semantic occupancy. The tokenization
and diffusion-based generative modeling is performed in this HexPlane latent
space.

At generation time, the diffusion transformer (DiT) component generates new
scenes by performing diffusion in the HexPlane space, optionally conditioning
the generation on a semantic map, or on past generation results to achieve
autoregressive synthesis. These samples then get decoded into 4D semantic
occupancy using the method described above.

The authors compare the approach to OccSora, another recent 4D generative
modeling technique, and show improved results on synthetic (CARLA-based) and
real (Waymo- and nuScenes-based) datasets.

**Strengths:**

- [S0] Flexible modeling approach which seems to scale well to large scenes while
  still allowing a wide range of rich conditioning methods.
- [S1] The proposed approach outperforms OccSora, a very modern competitor, in a
  wide range of metrics, including FID, precision, and recall.
- [S2] Some interesting implementation tricks could potentially be applied to
  other related tasks. For example, diffusion runs on a 2D setting with a clever
  tiling of the six HexPlanes into a single plane (Fig 4 - the "Padded Rollout").
- [S3] Overall, the paper is very well-written and provides thorough experiments,
  architectural details, and discussions. The appendix is likewise
  well-structured and I found it very easy to navigate.

**Weaknesses:**

- [W0] The pretrained networks used to calculate IS, FID, and KID should be
  motivated more thoroughly, especially in the 2D case.
  - For example, it is not clear why it is meaningful to use a CNN presumably
    trained on ImageNet or COCO to reason about samples consisting of semantic
    color maps. Unless this 2D CNN is trained to process semantic color maps as
    inputs, passing semantic color maps to such a CNN would produce OOD feature
    maps.
- [W1] One conceptual limitation is the fact that the method does not explicitly
  model uncertainty when forecasting a future scene conditioned on the present.
  Is this something that can be modeled by sampling multiple futures from the
  latent space?
- [W2] The core applications of 4D world modeling are tasks like simulation and
  motion forecasting. Presenting some results in this area could strengthen the
  paper. For example, this could include demonstrating that the world model
  performs well on a motion forecasting benchmark, or that it can be used to
  supplement training data for an end-to-end autonomous driving model.
- [W3] No source code seems to be promised.
- [W4] In the current stage, the approach is LiDAR-only. This is a minor
  limitation, though, and I am primarily mentioning it for completeness.
- Minor suggestions:
  - L245: "first generate" -> "first generates"
  - L866: Missing parentheses around the PyTorch citation.

**Questions:**

- [Q0] How do you calculate metrics like FID in 2D? What specific features do
  you use for the computation? BEV? What is the network used in these metrics
  originally trained on? I could not find this info in the references provided
  at the end of Section 5.1. If the pre-trained InceptionV3 and VGG-16 networks
  from A.2. are pre-trained on natural images, are they a good fit for comparing
  (what I assume to be) BEV semantic images?

---

> ### Author Response · Authors · 2024-11-23
> **Response to Reviewer nfdL**
>
> We sincerely thank Reviewer `nfdL` for the valuable comments provided during this review. The point-to-point responses are as follows.
>
> ---
>
> > **W0:** *"The pretrained networks used to calculate IS, FID, and KID should be motivated more thoroughly, especially in the 2D case ... Unless this 2D CNN is trained to process semantic color maps as inputs ..."*
>
> **A:** Thanks for your suggestion. We would like to clarify that the backbone used to calculate metrics in 3D is pretrained using occupancy data. In the 2D case, SemCity and OccSora use metrics such as IS, FID, and KID, but they do not provide any details of the pretrained networks they use. We use ImageNet-pretrained networks, following the common practice of most 2D diffusion methods. To alleviate your concern, we train InceptionV3 and VGG16 using rendered semantic color maps and re-evaluate our preformance. The evaluation results are provided below and have been updated in Table 2 of our paper.
>
> |Dataset|Method|IS $\uparrow$|FID $\downarrow$|KID $\downarrow$|P $\uparrow$|R $\uparrow$|
> |:-:|:-:|:-:|:-:|:-:|:-:|:-:|
> |CarlaSC|OccSora|1.030|28.549|0.008|0.224|0.010|
> |CarlaSC|Ours|1.040|12.936|0.002|0.307|0.018|
> |Occ3D-Waymo|OccSora|1.005|42.525|0.049| 0.654|0.004|
> |Occ3D-Waymo|Ours|1.010|36.727|0.001 | 0.705 | 0.015 |
>
> ---
>
> > **W1:** *"One conceptual limitation is the fact that the method does not explicitly model uncertainty when forecasting a future scene conditioned on the present. Is this something that can be modeled by sampling multiple futures from the latent space?"*
>
> **A:** Thank you for asking, but we are not sure about the "uncertainty" you mentioned. We assume it refers to the uncertainty in the generation process. Since we are using diffusion, it models generation uncertainty by randomly sampling from gaussian distribution for generation. In other words, different samples can lead to different generations, ensuring the generation uncertainty.
>
> ---
>
> > **W2:** *"... simulation and motion forecasting. Presenting some results in this area could strengthen the paper..."*
>
> **A:** Thank you for suggesting. This work mainly focuses on generating 4D scenes, which is an early study in this field, holding potentials for various downstream tasks. To address your concern, we conduct an experiment on occupancy forecasting as a comparison with OccWorld. Occupancy forecasting takes 2 seconds of occupancy context as input and predicts the next few seconds. The results are provided in the table below. Our method outperforms the previous state-of-the-art in most metrics.
>
> |Method|mIoU T=0|mIoU T=1|mIoU T=2|IoU T=0|IoU T=1| IoU T=2|
> |:-:|:-:|:-:|:-:|:-:|:-:|:-:|
> |OccWorld-O|66.38|25.78|15.14|62.29|34.63|25.07|
> |Ours|80.52|26.18|16.94|67.64|34.12|25.82|
>
> More details can be found in Appendix Section B.2.
>
> ---
>
> > **W3:** *"No source code seems to be promised."*
>
> **A:** Thanks for your comment. We commit to release our source code. We will update the codebase soon.
>
> ---
>
> > **W4:** *"In the current stage, the approach is LiDAR-only. This is a minor limitation, though, and I am primarily mentioning it for completeness."*
>
> **A:** Thanks for your suggestion. Our work mainly focus on generating 4D occumancy, which is challenging but holds the potential of various downstream tasks. We hope our work can inspire future studies involving additional modalities.
>
> ---
>
> > **W5:** *"Minor suggestions: L245: "first generate" -> "first generates"; L866: Missing parentheses around the PyTorch citation."*
>
> **A:** Thank you for pointing out the typos in our manuscript. We have corrected them in the updated paper.
>
> ---
>
> > **Q0:** *"How do you calculate metrics like FID in 2D? What specific features do you use for the computation? BEV? What is the network used in these metrics originally trained on? ... If ... are pre-trained on natural images, are they a good fit for comparing ... semantic images?"*
>
> **A:** Thanks for your question.
>
> - **2D Metrics.** We follow SemCity and use [torch-fidelity](https://github.com/toshas/torch-fidelity) to calculate the 2D metrics.
> - **Feature.** We render our features using a 45-degree angle perspective with the xy-plane above the scene instead of the BEV perspective (it lacks z-axis information). Reference images illustrating these viewpoints have been added to our [website](https://dynamic-city.github.io/#8-2d-evaluation-example) and our Appendix A.2 for clarity.
> - **Network.** We use ImageNet pre-trained networks for evaluation. Based on the feedback, we have fine-tuned the feature extractor specifically for semantic color maps and updated the results in our paper. The revised metrics can be found in Table 1 above.
>
> ---
>
> Last but not least, we sincerely thank Reviewer `nfdL` again for the valuable comments provided during this review.

---

> > ### Author Response · Authors · 2024-11-25
> > **Looking forward to hearing from you**
> >
> > **Dear Reviewer `nfdL`,**
> >
> > We sincerely appreciate your time and effort in reviewing our manuscript and providing valuable feedback.
> >
> > ---
> >
> > In response to your insightful comments, we have made the following revisions to our manuscript and project page:
> >
> > - We have changed the "LiDAR" in our title to "Occupancy" and added additional explanations to avoid ambiguity.
> > - We have updated the 2D generation evaluation results in Table 2 and included an example of the 2D semantic map in Appendix Sec. A.2.
> > - We have added the occupancy forecasting result in Appendix Sec. B.2.
> > - We have added user study results, including the evaluation of temporal consistency, in Appendix Sec. B.3.
> > - We have added a comparison of model speed, memory consumption, and size with OccSora in Appendix Sec. B.4.
> > - We have included a comparison with SemCity in Appendix Sec. B.5.
> > - We have added outpainting results in Appendix Sec. C.7.
> > - We have added single-frame occupancy conditional generation results in Appendix Sec. C.8.
> > - We have fixed typos and formatting errors.
> >
> > We hope these revisions adequately address your concerns.
> >
> > ---
> >
> > We look forward to actively participating in the **Author-Reviewer Discussion** session and welcome any additional feedback you might have on the manuscript or the changes we have made.
> >
> > ---
> >
> > Once again, we sincerely thank you for your contributions to improving the quality of this work.
> >
> > Best regards,
> >
> > The Authors of Submission 552

---

> > > ### Author Response · Authors · 2024-11-28
> > >
> > > **Dear Reviewer `nfdL`,**
> > >
> > > We sincerely thank you again for recognizing the contributions of our work and for your time and effort in reviewing our submission.
> > >
> > > Best regards,
> > >
> > > The Authors of Submission 552

---

### Official Review · Reviewer_D4sh · 2024-11-06

**Soundness:** 3
**Presentation:** 4
**Contribution:** 3
**Rating:** 8
**Confidence:** 4

**Summary:**

This paper proposes DynamicCity, a novel 4D LiDAR scene generation framework that supports large-scale dynamic reconstruction and generation. It introduces HexPlane as the compact 4D representation with effective decomposition to enhance the reconstruction quality. In order to improve the query efficiency, the authors further employ an expansion & squeeze strategy (ESS) to decode features in parallel. During the generation stage, this paper proposes a padded rollout operation to reorganize the six feature planes into a square feature map for better spatial and temporal awareness. Based on the VAE and DiT pipeline, DynamicCity achieves leading performance on both 4D reconstruction and generation, which also enables long sequential modeling and diverse conditional generation.

**Strengths:**

- Compared to existing methods that lack the ability of long-term dynamic generation, this paper utilizes Hexplane as the compact 4D representation and reorganizes into one feature map to achieve efficient reconstruction and generation.

- The decoding manner in parallel proposed in Expansion & Squeeze Strategy (ESS) alleviates the problem of dense queries and further improves the generation efficiency.

- Based on the VAE and DiT pipeline, the authors introduce diverse conditions for generation (e.g., command, trajectory and layout), demonstrating the potential of the model and its broad applications.

- The overall paper is easy to follow with excellent illustration and clear statements of contributions, making it very comfortable to read.

**Weaknesses:**

- Despite the compact HexPlane and parallel decoder, the dense feature volume and projection module of autoencoder are still very heavy, which limits its efficiency and scalability.

- The sample of the dataset is quite limited, which may lead to overfitting and memorization of the data by the generation model. This paper also lacks clarification of the division of training and test sets, as well as experiments and comparative results for their generalization ability and generative diversity.

- Although the authors provide diverse control conditions for generation, there is a lack of some more simple and practical conditions such as images, text or single-frame point clouds that are easily accessible.

**Questions:**

According to the weaknesses above, there are some concerns to be addressed:

1. It's noted that there are some recent works like XCube using advanced 3D sparse structure to improve the efficiency. Thus, it needs more explanation and comparison for the 3D backbone.

2. Does the model have the generalization ability and generative diversity? Overfitting to a few samples may reduce its significance. It would be better to provide more (conditional) generation results on the test set and multiple sampling results.

3. It would be much more beneficial to incorporate more practical conditions such as images, text or single-frame point clouds, which is also helpful for the assessment of generalization ability.

4. Given that the title is LARGE-SCALE LIDAR GENERATION, it may be plausible to include the generation or simulation of LiDAR point clouds (beyond occupancy) in the application. Or to avoid ambiguity.

5. How to better demonstrate temporal consistency compared to baselines, which is difficult to reflect in current metrics?

---

> ### Author Response · Authors · 2024-11-23
> **Response to Reviewer D4sh (1/2)**
>
> We sincerely thank Reviewer `D4sh` for the valuable comments provided during this review. The point-to-point responses are as follows.
>
> ---
>
> > **W1:** *"... the dense feature volume and projection module of autoencoder are still very heavy, which limits its efficiency and scalability.*"
>
> **A:** Thanks for your comment. We would like to clarify that the memory used by dense feature volumes with 16 frames on CarlaSC is only 8MB, and our projection modules are relatively small, each having only 4,352 parameters. While this operation is heavier than the average pooling used in SemCity, it significantly improves model performance, as demonstrated by our ablation study.
>
> ---
>
> > **W2:** *"The sample of the dataset is quite limited, which may lead to overfitting and memorization ... lacks clarification of the division of training and test sets, as well as experiments and comparative results ...*"
>
> **A:** Thanks for your suggestion.
>
> - **"Overfitting & Memorization".** It is possible to observe the memorization issue due to the use of diffusion models. But we want to highlight that our model can generate novel 4D lidar sequences, especially when we are performing conditional generations, even if we are using a relative small dataset for training the model.
> - **"Division of Training & Test Sets."** We use the default training and testing splits on CarlaSC, following the setup in SemCity. For Occ3D-Waymo, we use the validation split for testing, as the test split is not provided with the dataset.
> - **"Lack Experiments."** More generation results, especially conditional generation results, have been added to our [website](https://dynamic-city.github.io/) to prove our generalization ability and generative diversity. We would like to highlight that our method can perform conditional generations on HexPlane, command, and layout, which are not possible with previous works such as OccSora.
>
> ---
>
> > **W3:** *"... a lack of some more simple and practical conditions such as images, text or single-frame point clouds that are easily accessible."*
>
> **A:** Thank you for your constructive recommendation. We provide single-frame point cloud conditional generation result on our [website](https://dynamic-city.github.io/#7-single-occupancy-conditional-generation). We fine-tune our HexPlane conditional generation model to condition on single-frame point clouds. This is achieved by encoding a static scene as a HexPlane condition and training our HexPlane conditional generation pipeline as-is. Results can also be found in Appendix Section C.8.
>
> ---
>
> > **Q1:** *"... XCube using advanced 3D sparse structure ... more explanation and comparison for the 3D backbone."*
>
> **A:** We appreciate the mention of XCube, which uses sparse 3D convolution in its encoder and dense 3D convolution in its decoder. Our encoder uses a light 3D convolution before the main Projection Module and dense 3D convolution in the decoder. The use of advanced 3D sparse structures provides limited benefits to our autoencoder, as the encoder does not utilize many convolution operations. We plan to explore more efficient operations in the future to further optimize our 3D backbone. More discussion can be found in Appendix Section A.3.
>
> ---
>
> > **Q2:** *"Does the model have the generalization ability and generative diversity? ... provide more (conditional) generation results..."*
>
> **A:** Thanks for asking. Yes, our model could demonstrate strong generalization ability and generative diversity. More conditional generation results have been uploaded to our [website](https://dynamic-city.github.io/) to demonstrate that our model can be controlled to generate novel scenes.
>
> ---
>
> > **Q3:** *"... incorporate more practical conditions such as images, text or single-frame point clouds..."*
>
> **A:** Thank you for your advice. We have included single-frame point cloud conditional generation result in our [website](https://dynamic-city.github.io/#7-single-occupancy-conditional-generation). We also include the result in our Appendix Section C.8.
>
> ---
>
> > **Q4:** *"Given that the title is LARGE-SCALE LIDAR GENERATION ... Or to avoid ambiguity."*
>
> **A:** Thank you for pointing out this. To avoid ambiguity, we include necessary explanations in our paper, and we replace the term "LiDAR" with "Occupancy" in our title. In fact, occupancy represents concatenated dense LiDAR scenes.

---

> > ### Author Response · Authors · 2024-11-23
> > **Response to Reviewer D4sh (2/2)**
> >
> > > **Q5:** *"How to better demonstrate temporal consistency compared to baselines, which is difficult to reflect in current metrics?"*
> >
> > **A:** To better demonstrate temporal consistency compared to baselines, we designed a user study to assess temporal consistency along with several related aspects. The evaluation criteria includes: 1. overall quality; 2. temporal consistency; 3. background quality; 4. foreground quality.
> >
> > Our user study includes 20 samples in total, with 10 samples for each method. Each rating is on a scale of 1–5, where 1 indicates poor and 5 indicates excellent. We collecte responses from 42 volunteers and get 840 valid scores. The results are reported in the table below, and further details of the user study are provided in Appendix B.3.
> >
> > |Method|Overall Quality|Time Consistency|Background Quality|Foreground Quality|
> > |:-:|:-:|:-:|:-:|:-:|
> > |OccSora|2.21|2.05|2.17|2.11|
> > |Ours|4.03|4.02|3.95|4.04|
> >
> > ---
> >
> > Last but not least, we sincerely thank Reviewer `D4sh` again for the valuable comments provided during this review.

---

> > > ### Author Response · Authors · 2024-11-25
> > > **Looking forward to hearing from you**
> > >
> > > **Dear Reviewer `D4sh`,**
> > >
> > > We sincerely appreciate your time and effort in reviewing our manuscript and providing valuable feedback.
> > >
> > > ---
> > >
> > > In response to your insightful comments, we have made the following revisions to our manuscript and project page:
> > >
> > > - We have changed the "LiDAR" in our title to "Occupancy" and added additional explanations to avoid ambiguity.
> > > - We have updated the 2D generation evaluation results in Table 2 and included an example of the 2D semantic map in Appendix Sec. A.2.
> > > - We have added the occupancy forecasting result in Appendix Sec. B.2.
> > > - We have added user study results, including the evaluation of temporal consistency, in Appendix Sec. B.3.
> > > - We have added a comparison of model speed, memory consumption, and size with OccSora in Appendix Sec. B.4.
> > > - We have included a comparison with SemCity in Appendix Sec. B.5.
> > > - We have added outpainting results in Appendix Sec. C.7.
> > > - We have added single-frame occupancy conditional generation results in Appendix Sec. C.8.
> > > - We have fixed typos and formatting errors.
> > >
> > > We hope these revisions adequately address your concerns.
> > >
> > > ---
> > >
> > > We look forward to actively participating in the **Author-Reviewer Discussion** session and welcome any additional feedback you might have on the manuscript or the changes we have made.
> > >
> > > ---
> > >
> > > Once again, we sincerely thank you for your contributions to improving the quality of this work.
> > >
> > > Best regards,
> > >
> > > The Authors of Submission 552

---

> > > > ### Comment · Reviewer_D4sh · 2024-11-25
> > > >
> > > > Considering the effort of the authors during the rebuttal, which addressed most of my concerns, I will keep my positive score.

---

> > > > > ### Author Response · Authors · 2024-11-25
> > > > >
> > > > > **Dear Reviewer `D4sh`,**
> > > > >
> > > > > We are truly delighted to have addressed your concerns. Thank you again for recognizing the contributions of our work and for your time and effort in reviewing our submission.

---

### Author Response · Authors · 2024-11-23
**General Response**

We sincerely thank our reviewers for their time and effort in providing thorough and constructive feedback on our paper.

---

We are encouraged to see that our reviewers recognize this work as a novel and high-quality framework for 4D occupancy generation:

- Reviewers `D4sh`, `nfdL`, `a6Q2`, and `URnX` think our paper is *"well-written, easy to follow, and effectively structured"*.
- Reviewers `D4sh`, `nfdL`, and `a6Q2` think our pipeline is *"novel"* and *"introduces effective techniques such as Padded Rollout and ESS"*.
- Reviewers `D4sh`, `nfdL`, and `URnX` think our method *"achieves strong performance, outperforming baselines like OccSora in multiple metrics"*.
- Reviewers `D4sh`, `a6Q2`, and `URnX` think our work *"demonstrates broad applicability, supporting diverse conditional generation tasks"*.

---

We highlight our contributions as follows:

- **DynamicCity** encodes 4D Occupancy as HexPlanes using a novel VAE that incorporates **Projection Modules** and an **Expansion & Squeeze Strategy (ESS)**. Additionally, we propose a **Padded Rollout Operation** on the fitted HexPlanes, enabling the modeling of spatial and temporal relationships during the diffusion process. Finally, we showcase a variety of **applications** to demonstrate the potential of our models.

---

As suggested by the reviewers, we revised our manuscript and website as follows:

***Additional Results:***
- Updated the 2D generation evaluation results in Table 2 and included an example of the 2D semantic map in Appendix Section A.2, as suggested by Reviewer `nfdL`.
- Added occupancy forecasting result, as suggested by Reviewer `nfdL` in Appendix Section B.2.
- Added user study results, including the evaluation of temporal consistency, in the Appendix Section B.3 as suggested by Reviewer `D4sh`.
- Added a comparison of model speed, memory consumption, and size with OccSora in the Appendix Section B.4, as suggested by Reviewers `a6Q2` and `URnX`.
- Included a comparison with SemCity in the Appendix Section B.5, as suggested by Reviewer `a6Q2`.
- Added outpainting results in Appendix Section C.7 as suggested by Reviewer `a6Q2`.
- Added single-frame occupancy conditional generation results in Appendix Section C.8 as suggested by Reviewer `D4sh`.

***Other Modifications:***
- Changed the "LiDAR" in our title to "Occupancy" and added additional explanations to avoid ambiguity, as suggested by Reviewers `D4sh`, `a6Q2`, and `URnX`.
- Fixed typos and formatting errors, as suggested by Reviewer `nfdL`.

---

### Meta-Review · Area_Chair_csJe · 2024-12-22

**Metareview:**

The paper introduces DynamicCity, a novel framework for 4D LiDAR scene generation. The proposed method is enabled by careful integration of a set of features: (i) HexPlane representation, (ii) Expansion & Squeeze Strategy (a parallel decoding mechanism that addresses dense queries, enhancing query efficiency and scalability), (iii) Padded Rollout Operation, and (iv) Diffusion-based Generation. While the reviewers initially have some concerns about efficiency, generalization, and metrics, the authors carefully addressed most of them during the rebuttal discussion phase. The paper, at the end, received unanimous positive reviews. The ACs agreed with the reviewers that the execution of the paper is extraordinary and the proposed method is very effective in addressing 4D scene generation. It is exciting to see that the introduced method supports diverse conditional generation methods and can model sequences up to 128 frames. It is a clear accept.

**Additional Comments On Reviewer Discussion:**

The authors addressed most concerns during the rebuttal phase. Good job!

---

### Decision · Program_Chairs · 2025-01-22

Accept (Spotlight)